# Micronucleus is not a potent inducer of the cGAS/STING pathway

Yuki Sato[1,2], Makoto T Hayashi[2,3]

**Micronuclei (MN) have been associated with the innate immune response. The abrupt rupture of MN membranes results in the accumulation of cGAS, potentially activating STING and downstream interferon-responsive genes. However, direct evidence connecting MN and cGAS activation has been lacking. We have developed the FuVis2 reporter system, which enables the visualization of the cell nucleus carrying a single sister chromatid fusion and, consequently, MN. Using this FuVis2 reporter equipped with cGAS and STING reporters, we rigorously assessed the potency of cGAS activation by MN in individual living cells. Our findings reveal that cGAS localization to membrane-ruptured MN during interphase is infrequent, with cGAS primarily capturing MN during mitosis and remaining bound to cytosolic chromatin. We found that cGAS accumulation during mitosis neither activates STING in the subsequent interphase nor triggers the interferon response. Gamma-ray irradiation activates STING independently of MN formation and cGAS localization to MN. These results suggest that cGAS accumulation in cytosolic MN is not a robust indicator of its activation and that MN are not the primary trigger of the cGAS/STING pathway.**

## Introduction

Micronuclei (MN), small chromatin-containing compartments in the cytosol, are isolated from the primary nucleus (PN) and are frequently observed in aging tumor cells, and cells exposed to genotoxic insults. Consequently, MN serve as a reliable biomarker for chromosome instability (Krupina et al, 2021). MN can form as a result of chromosome missegregation because of lagging chromosomes, acentric chromosome fragments (Fenech et al, 2011; Thompson & Compton, 2011), and breakage of anaphase chromatin bridges (Kagaya et al, 2020; Umbreit et al, 2020). The genetic material in MN undergoes dysregulated DNA replication and DNA damage repair (Crasta et al, 2012), potentially leading to chromothripsis events (Zhang et al, 2015; Ly et al, 2017, 2019; Kneissig et al, 2019;

Umbreit et al, 2020). Recently, MN have been associated with the activation of the innate immune response through the cyclic GMP-AMP synthase (cGAS) and the stimulator of interferon genes (STING) pathway (Dou et al, 2017; Glück et al, 2017; Harding et al, 2017; Mackenzie et al, 2017).

cGAS is activated by a cytosolic double-stranded DNA, resulting in the production of the second messenger 2'3'-cyclic GMP-AMP (cGAMP). cGAMP is detected by STING, leading to its activation through translocation from the endoplasmic reticulum (ER) to the ER–Golgi intermediate compartment and the Golgi apparatus (Hopfner & Hornung, 2020). STING subsequently activates TANK-binding kinase 1 (TBK1), which then phosphorylates itself, STING, and the interferon regulatory factor 3 (IRF3) transcription factor. This cascade promotes the translocation of IRF3 into the nucleus, ultimately resulting in the activation of type I interferons and interferon-stimulated genes (ISGs) (Hopfner & Hornung, 2020). STING also exhibits interferon-independent activity through the TBK1-dependent I$\kappa$B kinase $\varepsilon$ (IKK$\varepsilon$) recruitment and downstream NF-$\kappa$B response (Balka et al, 2020), as well as cGAS-independent non-canonical activity upon DNA damage that does not involve translocation to the Golgi (Dunphy et al, 2018). Although cGAS was initially reported to reside in the cytosol to prevent self-DNA activation (Wu et al, 2013), recent studies revealed that cGAS is present not only in the cytosol (Barnett et al, 2019) but also in the nucleus during interphase (Yang et al, 2017; Gentili et al, 2019), and accumulates on mitotic chromosomes (Harding et al, 2017; Yang et al, 2017; Gentili et al, 2019; Zierhut et al, 2019). Cryo-EM structures of the cGAS–nucleosome complex have demonstrated that the interaction between cGAS and histone H2A-H2B dimers sequesters the DNA-binding site of cGAS required for activation (Boyer et al, 2020; Cao et al, 2020; Kujirai et al, 2020; Michalski et al, 2020; Zhao et al, 2020). In addition, during mitosis, hyperphosphorylation of the N-terminal disordered region of cGAS has been shown to inhibit its activation (Li et al, 2021).

It has been proposed that the nuclear membrane of MN ruptures during interphase, enabling the activation of cGAS by MN (Dou et al, 2017; Glück et al, 2017; Harding et al, 2017; Mackenzie et al, 2017; Yang et al, 2017). However, these studies mostly relied on cell populations to analyze cGAS localization to MN and cGAS/STING-dependent

[1]Graduate School of Biostudies, Kyoto University, Kyoto, Japan    [2]IFOM-KU Joint Research Laboratory, Graduate School of Medicine, Kyoto University, Kyoto, Japan    [3]IFOM ETS, the AIRC Institute of Molecular Oncology, Milan, Italy

Correspondence: hayashi.makoto.8a@kyoto-u.jp; makoto.hayashi@ifom.eu

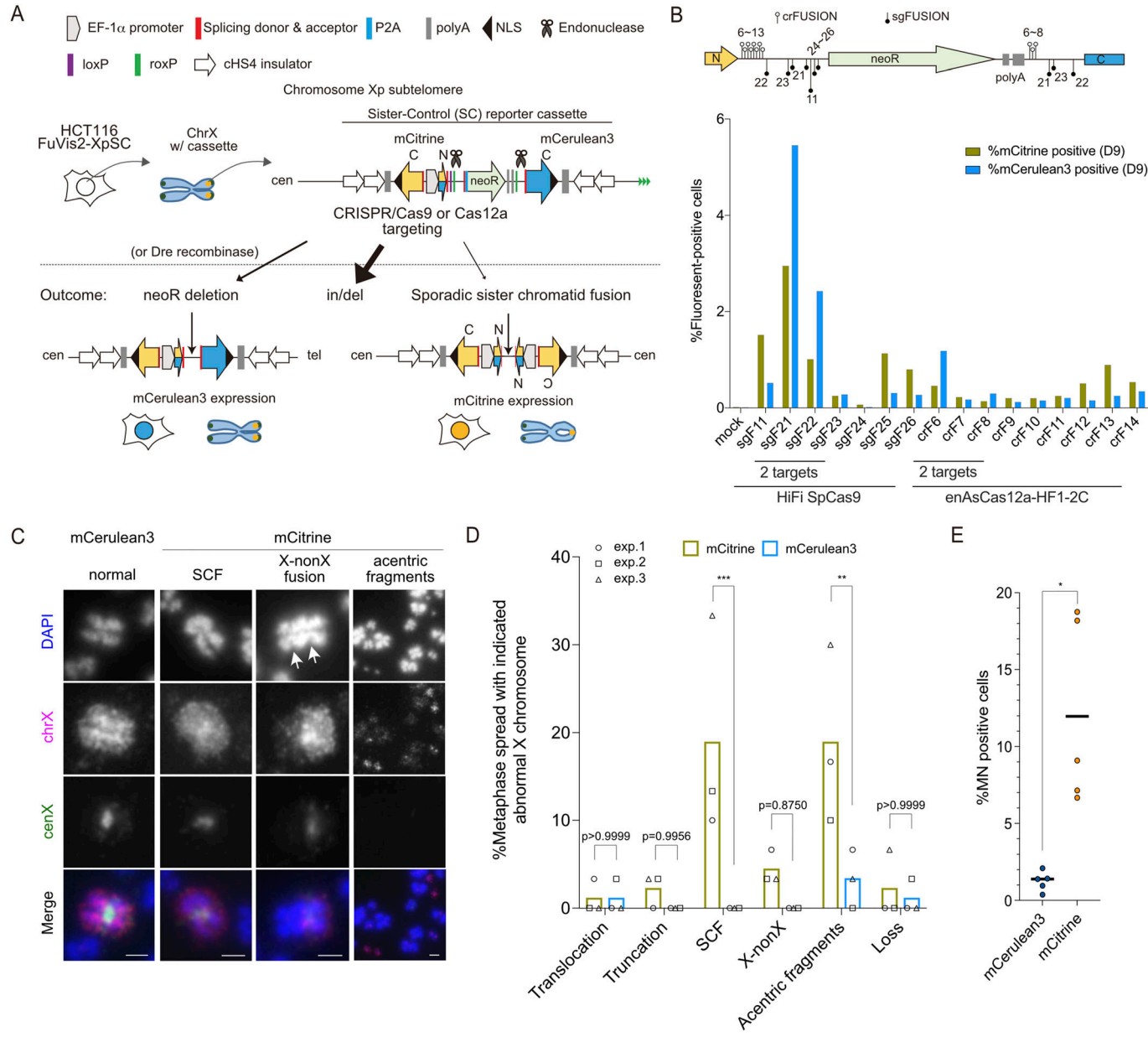

**Figure 1. Validation of the FuVis2-XpSC reporter system.**
**(A)** Schematic of the FuVis2 reporter system in HCT116 cells. Integration of the Sister-Control (SC) reporter cassette into X chromosome subtelomere, and CRISPR/endonuclease targeting outcomes, including neoR deletion (mCerulean3 expression) and sister chromatid fusion (mCitrine expression). **(B)** mCitrine- and mCerulean3-positive XpSC cell percentages with various endonucleases and guide RNAs at 9 d post-infection. Guide RNA target sequence positions are shown above. **(C)** Representative images of the X chromosome in XpSC33 Cas9-sgF21 cells, post-sorting and chromosome spread using chrX and cenX probes. Arrows indicate centromere loci. Parts of whole spreads in Fig S1F are shown. Scale bars: 2 μm. **(D)** Quantification of X chromosome abnormalities from (C) (n = 30/experiment; three biological replicates). **(E)** MN-positive cell percentages in XpSC33 Cas9-sgF21 cells 6 d post-transduction (n ≥ 15/experiment for mCitrine and ≥ 216/experiment for mCerulean3; five biological replicates). Data information: **(D)** bars represent the mean, **P < 0.01 and ***P < 0.001 (ordinary one-way ANOVA followed by Sidak's multiple comparison). **(E)** Bars represent the mean, *P < 0.05 (Welch's t test).
Source data are available for this figure.

interferon responses, lacking direct evidence that MN activate cGAS/STING in the same cell. This raises the question of how cGAS can be efficiently activated by MN in the presence of suppressive chromatin–cGAS interaction, with a recent study suggesting that MN may not activate cGAS (Flynn et al, 2021). Notably, irradiation, commonly used to induce MN, has been shown to

cause mitochondrial DNA (mtDNA) damage and a mitochondria-dependent innate immune response (Tigano et al, 2021). These findings raise the possibility that severe genotoxic insults leading to both MN formation and mitochondrial damage may trigger mtDNA-dependent cGAS activation (Kim et al, 2023). To address whether micronucleus is a potent activator of cGAS, a reporter

system capable of inducing MN without affecting mitochondrial integrity and enabling the tracking of MN formation, cGAS localization, and STING activation in live cells is required.

We previously developed a cell-based reporter system known as the Fusion Visualization (FuVis) system, which allows for the visualization of cells with defined single sister chromatid fusions (SCF) (Kagaya et al, 2020). Live-cell imaging demonstrated that the most prominent phenotype resulting from SCF is MN formation in subsequent cell cycles (Kagaya et al, 2020). Given that the MN induced in the FuVis system originate solely from anaphase chromatin bridges caused by SCF, the FuVis reporter provides a unique system to study cGAS/STING activity upon MN formation without affecting the mitochondrial function.

# Results

## Second generation of the Fusion Visualization system

The first generation of the FuVis reporter system (FuVis1) comprised two distinct cell lines: FuVis-XpSIS and FuVis-XpCTRL. Both cell lines contained integrated artificial cassette sequences near telomeres on the short arm of the X chromosome, incorporating two exons (154 and 563 bp) of the mCitrine gene in different configurations, allowing for the detection of SCF (XpSIS) or DNA damage repair without SCF (XpCTRL) through mCitrine expression (Kagaya et al, 2020). Notably, these cell lines exhibited slight variations in morphology and growth rates, indicating potential genetic or epigenetic differences arising during the cloning process, which presented challenges in interpreting the precise effects of SCF (Kagaya et al, 2020). In response to this limitation, we aimed to develop an improved FuVis system capable of detecting both SCF and DNA damage repair distinctively in a single reporter cell line (Fig 1A). Taking advantage of the shared N-terminus amino acid sequences between mCitrine and mCerulean3, we inserted a corresponding 3′-exon of mCerulean3 downstream of the neomycin-resistance gene (neoR) and polyA sequences within the original sister cassette sequence (Fig 1A). By targeting spacer sequences flanking the neoR with RNA-guided endonucleases, we enabled neoR deletion, followed by mCerulean3 expression (Fig 1A, neoR deletion), as well as sporadic SCF, followed by mCitrine expression (Fig 1A, sporadic SCF). We successfully isolated a FuVis2-XpSC33 clone that harbors a single reporter cassette integration without apparent karyotypic or growth defects (Fig S1A–E; please refer to the Materials and Methods section for details).

To validate the FuVis2 reporter, we targeted various sequences flanking neoR using two endonucleases: the SpCas9 variant HiFi SpCas9 [Cas9(HiFi)] and AsCas12a variant enAsCas12a-HF1-2C [Cas12a(HF1)] (Vakulskas et al, 2018; Kleinstiver et al, 2019). Guide RNAs (sgFUSION and crFUSION) were designed for both endonucleases to target either a single site upstream of neoR or two sites flanking neoR (Fig 1B). XpSC33 cells were transduced with a virus encoding either Cas9(HiFi)-sgFUSION (sgF) or Cas12a(HF1)-crFUSION (crF) and analyzed on day 9 using flow cytometry. Among these constructs, only guide RNAs targeting two neoR-flanking sequences (sgF21, sgF22, and crF6) induced both mCerulean3 and

mCitrine expression (Fig 1B). Guide RNAs targeting a single site (sgF11, sgF25, sgF26, crF12, crF13, and crF14) induced mCitrine expression with a background level of mCerulean3 expression (Fig 1B). For subsequent analysis, we selected Cas9(HiFi)-sgF21 (hereafter Cas9-sgF21), which induced the highest levels of both mCitrine and mCerulean3.

To analyze X chromosome abnormalities, mCitrine- and mCerulean3-positive XpSC33 Cas9-sgF21 cells were sorted and subjected to dual-colored FISH analysis using whole X chromosome painting (chrX) and chromosome X centromere-specific (cenX) probes. Compared with mCerulean3-positive cells, mCitrine-positive cells exhibited a significantly increased rate of abnormal X chromosomes, including SCF and acentric fragments (Figs 1C and D and S1F and G). Although a slight increase in fusion between X and non-X chromosomes was also observed, it did not reach statistical significance (Fig 1C and D). Because SCF is typically only observed in the first mitosis after formation, not all mCitrine-positive mitotic cells displayed SCF. Although we cannot rule out other causes for mCitrine expression, acentric fragments and chromosome fusions likely arose as secondary abnormalities stemming from SCF breakage after the first mitosis. Time-course analysis of XpSC33 cells expressing different endonuclease and guide RNA pairs showed that mCerulean3-positive cells reached a plateau as early as 6 d post-infection, whereas mCitrine-positive cells peaked around day 6 and gradually decreased, irrespective of the efficiency of the endonucleases and guide RNAs used (Fig S1H). This kinetic pattern aligns with the assumption that a single mCitrine gene locus generated by SCF can be transmitted to either one of two daughter cells, resulting in the gradual loss of the mCitrine protein in the other lineage that did not inherit the mCitrine gene (Kagaya et al, 2020). Importantly, mCitrine-positive cells exhibited increased MN formation compared with mCerulean3-positive cells 6 d post-infection (Fig 1E). These findings are consistent with previous results obtained from the FuVis1 system, confirming that a single SCF event can lead to MN formation.

## SCF causes micronuclei after the first mitosis

To investigate the kinetics of MN formation in the FuVis2 system, we conducted live-cell imaging using XpSC33 Cas9-sgF21 cells. During the first interphase when cells became fluorescent, both mCitrine- and mCerulean3-positive cells displayed background levels of MN (6.7% and 7.6%, respectively) (Fig 2A–C). However, in the second interphase, 40.6% of mCitrine-positive cells developed MN, whereas mCerulean3-positive cells remained unchanged (6.0%) (Fig 2C). This result further supports the notion that MN originate from a single SCF event that experienced breakage during the first mitotic exit.

The continuous expression of Cas9 raises concerns about potential off-target genomic damage, which could lead to unintended MN formation. To address this concern, we isolated a clone of XpSC33 cells equipped with doxycycline (dox)-inducible Cas9(HiFi), subsequently renamed as XpSC33-iCas9-20 (Fig S2A–F; please refer to the Materials and Methods section for details). XpSC33-iCas9-20 cells were transduced by the sgF21-encoding virus in the presence of 0.1 µg/ml dox for 1 d and analyzed from days 2–6 using flow cytometry, confirming the

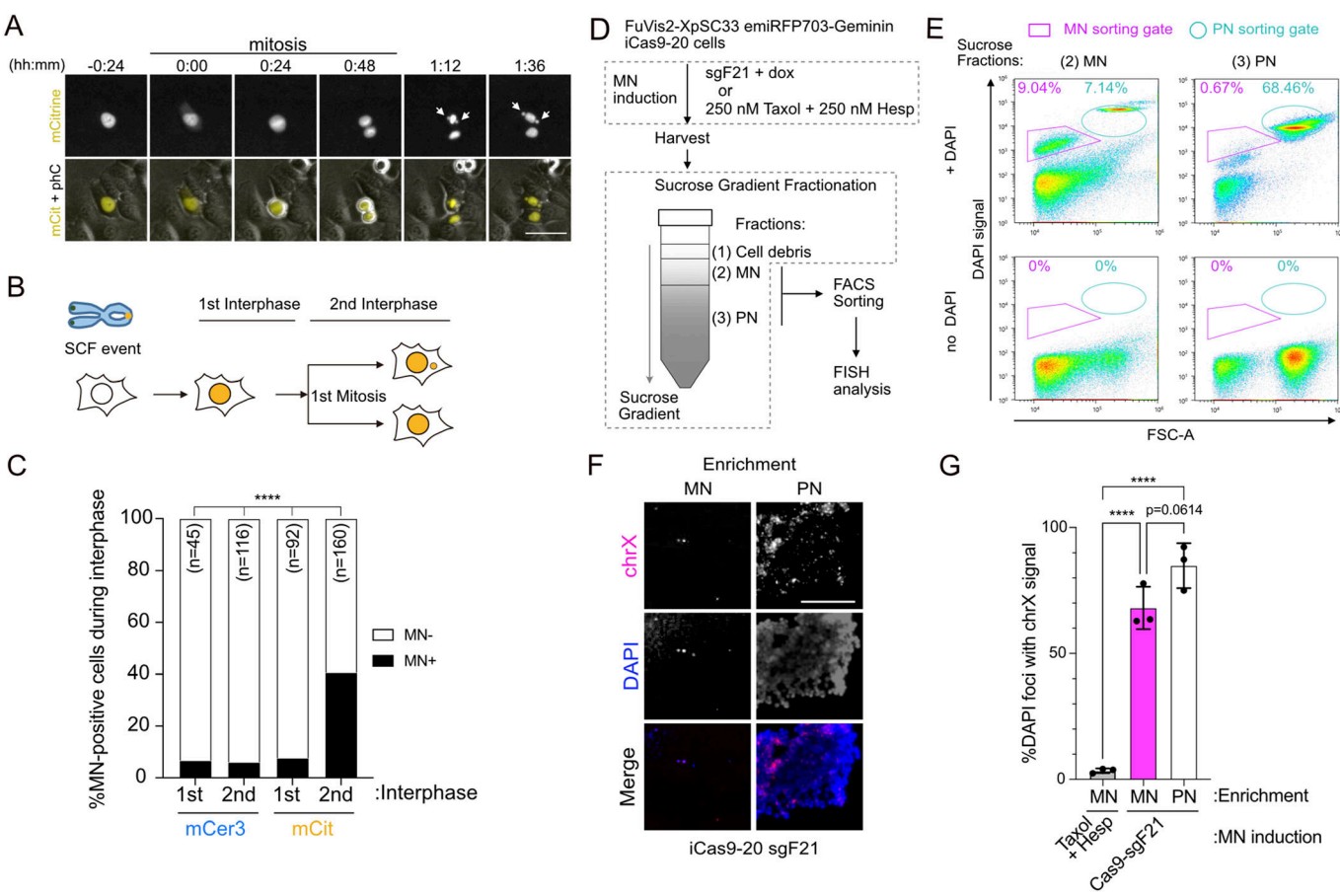

**Figure 2. Sister chromatid fusion causes micronuclei after the first mitosis.**
**(A)** Live-cell images of MN formation in mCitrine-positive XpSC33 Cas9-sgF21 cells. MN indicated by white arrows. Scale bar: 25 μm. phC: phase contrast. **(B)** Schematic of cell cycle progression post-SCF event. **(C)** MN-positive cell percentages at the indicated cell cycle stages in mCerulean3- and mCitrine-positive XpSC33 Cas9-sgF21 cells, analyzed from days 4–7 post-infection. **(D)** Method for MN and PN enrichment. XpSC33 emiRFP703-Geminin iCas9-20 cells were either transduced with sgF21 or exposed to Taxol and Hesperadin (250 nM each). Cell extracts were fractionated and sorted for MN and PN enrichment. **(E)** FACS analysis of DAPI-stained MN and PN. **(F)** FISH images of MN- and PN-enriched samples using a chrX probe. Results from XpSC33 emiRFP703-Geminin iCas9-20 sgF21 cells are shown. Scale bar: 10 μm. **(G)** Percentage of DAPI foci with chrX signals (n ≥ 70/experiment; three biological replicates). Data information: **(C)** ****P < 0.0001(chi-square test). **(G)** Mean ± SD, ****P < 0.0001 (ordinary one-way ANOVA followed by Tukey's multiple comparison).
Source data are available for this figure.

expected expression of both mCitrine and mCerulean3 (Fig S2G). Live-cell analysis revealed a significant increase in MN-positive cells during the second interphase among mCitrine-positive, but not mCerulean3-positive, cells (Fig S2H).

To further validate the nature of MN, we aimed to purify SCF-derived MN from XpSC33 cells. Because MN isolation requires a sufficient number of cells, and mCitrine-positive cells are rare, we decided to use the entire population of sgF21-expressing XpSC33 iCas9-20 cells. However, both mCitrine- and mCerulean3-positive populations exhibited MN-positive cells in the first interphase (Figs 2C and S2H), likely stemming from background MN formation unrelated to the SCF event. Because these cells do not divide frequently, we attempted to collect a cycling population to accumulate cells with SCF-derived MN. For this purpose, XpSC33 iCas9-20 cells were transduced with a virus encoding emiRFP703-Geminin, a derivative of the FUCCI reporter system for visualizing the S/G2/M phase of the cell cycle (Sakaue-Sawano et al, 2008). Transduced cells were sequentially sorted twice to enrich for cells

with the expected reporter expression, validated by aphidicolin treatment and serum starvation (Fig S2I). The resulting XpSC33 emiRFP703-Geminin iCas9-20 cells were transduced with the sgF21-encoding virus, and emiRFP703-positive cells were sorted on day 8 post-infection. Cell extracts were subjected to sucrose gradient fractionation and sorting by DAPI staining for MN and PN purification (Fig 2D and E). The resulting MN- and PN-enriched samples were subjected to FISH analysis using the chrX probe. As anticipated, the PN-enriched sample consistently exhibited chrX focus formation (Fig 2F and G). Remarkably, we found that the MN-enriched sample was very frequently painted with the chrX probe (Fig 2F and G). In contrast, a similar painting was not observed in an MN-enriched sample from cells treated with a microtubule stabilizer Taxol and Aurora kinase B inhibitor Hesperadin for 48 h (Fig 2D and G). Collectively, these results suggest that the SCF-derived chromatin bridge of X chromosomes is disrupted during the first mitosis, leading to MN formation in the subsequent cell cycle. Thus, the FuVis2 reporter system offers a unique

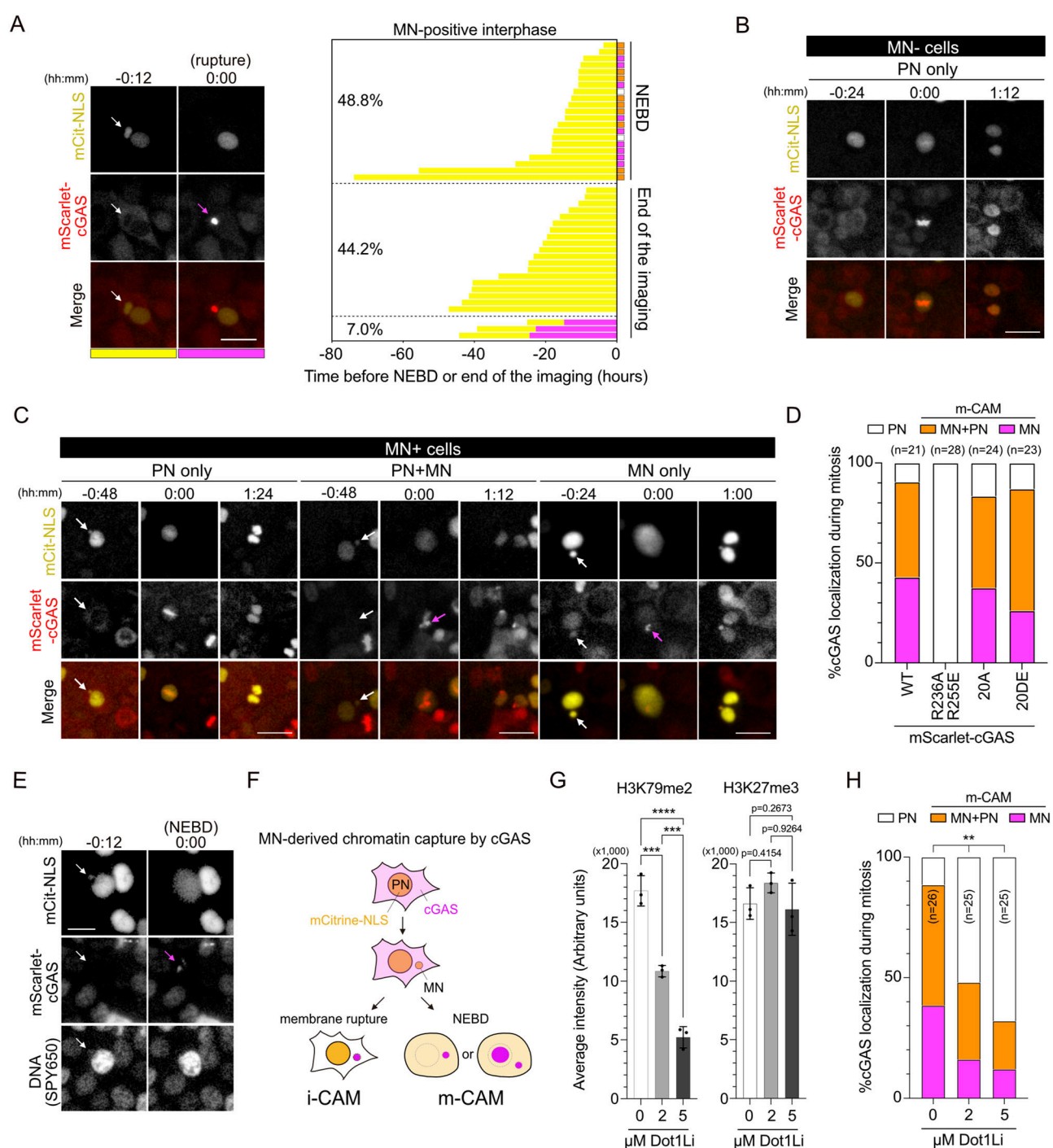

**Figure 3. SCF-derived MN is captured by cGAS upon mitotic nuclear envelope breakdown.**
**(A)** Live-cell analysis of MN captured by cGAS during interphase in XpSC33 mScarlet-cGAS Cas9-sgF21 cells. Left: MN rupture event (white arrows: intact MN; magenta arrow: cGAS accumulation). Scale bar: 25 μm. Right: cGAS localization in mCitrine- and MN-positive cells as they progress through interphase to mitosis (NEBD) or the end of imaging (set as T = 0), with color-coded bars representing the MN status. For the NEBD category, cGAS localization patterns in mitosis, as analyzed in (D), are also indicated. **(B, C)** Pre- and post-NEBD images of mCitrine-NLS and mScarlet-cGAS in MN-negative (B) and MN-positive (C) cells. NEBD indicated by mCitrine-NLS diffusion (0:00); arrows show intact MN (white) and cGAS foci on MN-derived chromatin upon NEBD (magenta). Scale bars: 25 μm. **(D)** Percentage of cGAS localization patterns upon NEBD in MN-positive cells with WT and mutant mScarlet-cGAS. **(E)** Live-cell images of mScarlet-cGAS at MN-derived DNA locations, as shown in (C). Scale bar: 10 μm. **(F)** Schematic illustrating two distinct pathways of cGAS in the initial capture of MN. **(G)** H3K79me2 and H3K27me3 signal intensities in XpSC33 mScarlet-cGAS cells exposed to SGC0946 for 1 wk (n ≥ 30/experiment; three biological replicates). **(H)** Percentage of cGAS localization patterns upon NEBD as shown in (D). Data information: **(G)** mean ± SD, ***$P$ < 0.001 and ****$P$ < 0.0001 (ordinary one-way ANOVA followed by Tukey's multiple comparison). **(H)** **$P$ < 0.01 (chi-square test).
Source data are available for this figure.

opportunity to explore the fate of MN originating solely from a single SCF event on the short arm of the X chromosome.

## MN-derived chromatin is captured by cGAS upon mitotic nuclear envelope breakdown (NEBD)

Previous studies have suggested that the MN membrane ruptures during interphase, leading to the accumulation and activation of cGAS (Dou et al, 2017; Harding et al, 2017; Mackenzie et al, 2017). We refer to this phenomenon as "interphase-cGAS accumulation in MN" or "i-CAM" and aimed to determine the frequency of i-CAM in XpSC33 Cas9-sgF21 cells expressing mScarlet-cGAS. We transduced XpSC33 cells with mScarlet-cGAS–encoding virus, sorted them three times, and confirmed mScarlet-cGAS expression (Fig S3A). A long-term live-cell analysis of mCitrine-positive cells revealed that i-CAM is a rare event, occurring in only 7.0% of MN-positive cells (Fig 3A). Instead, we observed unique cGAS localization patterns during mitosis, which could be classified into three categories. First, in MN-negative cells and 9.5% of MN-positive cells, mitotic cGAS localized to PN-derived chromosomes, consistent with previous reports (Harding et al, 2017; Gentili et al, 2019; Zierhut et al, 2019) (Fig 3B–D; PN only). Second, in 47.6% of MN-positive cells, cGAS localized to both MN- and PN-derived chromosomes (Fig 3C and D; PN+MN). Lastly, in 42.9% of MN-positive cells, cGAS robustly accumulated in the MN-derived chromosome region upon NEBD (Fig 3C–E; MN only). Collectively, these findings revealed that 90.5% of MN-positive cells that entered mitosis exhibited mitotic cGAS accumulation in MN-derived chromatin, which we term "m-CAM" (Fig 3F).

We further tracked the reformation of MN and cGAS localization in the subsequent interphase, categorizing them into four groups (Fig S3B): (1) mCitrine-positive MN with cGAS accumulation, (2) mCitrine-negative MN with cGAS accumulation, (3) mCitrine-positive MN without cGAS accumulation, and (4) no evidence of MN. Notably, we observed that cGAS accumulated in MN in approximately half of the m-CAM–derived MN-positive cells (groups 1, 2, and 3), which accounts for about 30% of the total m-CAM–derived population (Fig S3B). This observation aligns with previous reports demonstrating cGAS accumulation in MN among fixed interphase cells (Dou et al, 2017). Our results suggest that the cGAS accumulation in MN observed in fixed cells mainly arises from MN that have experienced the m-CAM event.

To explore the mechanism behind m-CAM, XpSC33 Cas9-sgF21 cells expressing three cGAS mutants were subjected to live-cell analysis. We discovered that the cGAS$^{R236A–R255E}$ mutant, which carries mutations on the nucleosome-binding surface (Volkman et al, 2019), completely abolished the m-CAM event while retaining localization to PN-derived chromosomes (Figs 3D and S3A and C). On the contrary, no effect on m-CAM was observed in cells expressing phosphomimetic (cGAS$^{20DE}$) and phospho-null (cGAS$^{20A}$) mutants of its N-terminal domain, which harbor mutations in 20 Ser/Thr sites required for mitotic inactivation of cGAS (Li et al, 2021) (Figs 3D and S3A and C). We confirmed that cGAS$^{20A}$ and cGAS$^{20DE}$ exhibit m-CAM even under the knockdown of endogenous CGAS (Fig S3D and E). These results indicate that the nucleosome-binding ability of cGAS is crucial for m-CAM, which is distinct from mitotic cGAS localization to PN-derived chromosomes. We further addressed whether m-CAM is influenced by modifying the histone

modification, H3K79me2, known to recruit cGAS to interphase MN (MacDonald et al, 2023). Pre-treatment with a DOT1L inhibitor SGC0946 for 7 d, which abolishes H3K79me2, but not H3K27me3 (MacDonald et al, 2023) (Figs 3G and S3F), significantly suppressed the m-CAM event (Fig 3H). These results suggest that the H3K79me2 mark on MN allows cGAS to interact more efficiently with nucleosomes upon mitotic entry.

## m-CAM does not lead to STING activation

The dominance of the m-CAM event and the persistence of cytoplasmic cGAS foci in the subsequent interphase raised the possibility that STING is activated in the subsequent cell cycle. Because TBK1 and IRF3 can be activated independently of cGAS/STING pathways (Liu et al, 2015), we aimed to directly monitor the activity of STING. To achieve this, XpSC33 cells were transduced with viruses encoding emiRFP703-cGAS and mRuby3-STING reporters (Balka et al, 2023; Kuchitsu et al, 2023). STING translocates from the ER to the Golgi apparatus during activation (Mukai et al, 2016). Consistently, mRuby3-STING accumulated at the Golgi apparatus 2 h after exposure to compound 3, a potent STING agonist (Ramanjulu et al, 2018) (Fig 4A). We used the maximum intensity and average intensity of mRuby3-STING in a cell to assess STING accumulation as an indicator of its activation (Fig 4A, STING Accumulation Index: St-AI). To validate the reliability of St-AI, cells were immunostained for pSTING-S366, a TBK1-dependent phosphorylation indicative of its activation (Liu et al, 2015) (Fig 4B). Based on the scatter plot of pSTING-S366 intensity and St-AI, we observed a strong correlation between St-AI values and pSTING-S366 signal intensities (Fig S4A). We defined St-AI values greater than 2.0 as indicative of STING activation (Figs 4C and S4A). Serial dilution of compound 3 showed that pSTING-S366 intensity and St-AI exhibited the same threshold concentration for indicating STING activation (Figs 4D and S4B and C), which correlated well with the up-regulation of CXCL10, an interferon gamma–induced inflammatory marker (Fig 4E). Time-lapse analysis confirmed that compared with the mock control, St-AI gradually increased after the transfection of pMAX-TurboGFP (GFP) plasmid as a source of cytosolic dsDNA (Figs 4F and G and S4D). shRNA knockdown of CGAS completely abolished the increase in St-AI after pMAX-GFP transfection but not compound 3 (Figs 4H and S4E and F), confirming cGAS-dependent STING activation in the presence of cytosolic dsDNA. The attenuation of St-AI by shcGAS under the compound 3 condition may be attributed to the loss of secondary activation of the cGAS/STING cycle caused by dsDNA released from dead cells (Messaoud-Nacer et al, 2022). In conclusion, we consider St-AI a valuable indicator of STING activation in live cells.

To address STING activation after m-CAM, we performed live-cell imaging in XpSC33 emiRFP703-cGAS mRuby3-STING cells transduced with the Cas9-sgF21–encoding virus. We first confirmed that lentivirus transduction itself does not activate STING (Fig S4G) and that the m-CAM event is dominant over the i-CAM event under these conditions as well (Fig S4H). Time-course analysis revealed that St-AI remained unchanged during the interphase after the m-CAM event (Fig 4I). Because both nucleosome binding and mitotic hyperphosphorylation attenuate

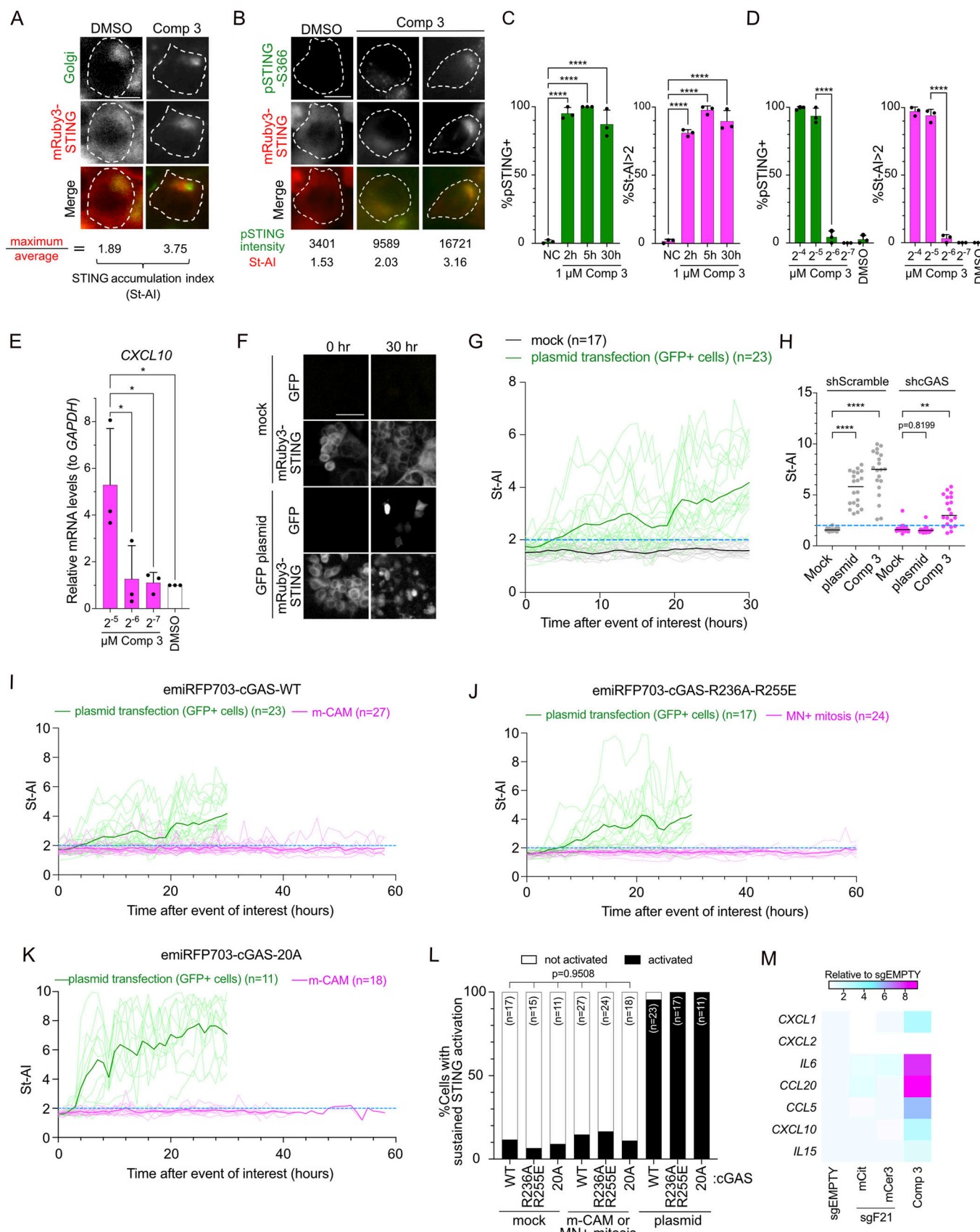

cGAS activation (Volkman et al, 2019; Li et al, 2021), we performed the same experiments in cells expressing emiRFP703-cGAS$^{R236A–R255E}$ and emiRFP703-cGAS$^{20A}$ (Fig 4J and K). To compare the STING activation rates under various live-cell imaging conditions, we defined sustained STING activation as St-AI values exceeding 2.0 for a duration over 4 h during a specified interphase. We found no significant increase in sustained STING activation after m-CAM, compared with the plasmid transfection control, in cells expressing not only cGAS-WT but also –R236A–R255E and –20A (Fig 4L). We attempted but failed to obtain XpSC33 cells expressing emiRFP703-cGAS$^{R236A–R255E–20A}$ mutant because of strong toxicity (Li et al, 2021). These results suggest that cGAS activation is strongly suppressed during and after the m-CAM event. In agreement with this result, neither mCitrine- nor mCerulean3-positive XpSC33 Cas9-sgF21 cells showed induction of ISGs (Figs 4M and S5A). Given that senescence can result from cGAS/STING activation (Glück et al, 2017), we isolated mCitrine- and mCerulean3-positive XpSC33 Cas9-sgF21 cells and recultured them for 10 d. These cells were assessed for senescence by checking *CDKN1A* induction and *LMNB1* reduction, established senescence markers (Shimi et al, 2011) (Fig S5B). Unlike bleomycin-induced senescent cells, mCitrine-positive cells showed no signs of senescence (Fig S5C). In conclusion, the data suggest a lack of cGAS activation in the mCitrine-positive population.

Previous research has shown that the TREX1 exonuclease localizes at MN, potentially inhibiting cGAS activation (Mohr et al, 2021). TREX1 staining in XpSC33 Cas9-sgF21 cells showed MN localization in mCitrine-positive cells (Fig S6A), suggesting a suppressive role of TREX1. To investigate this further, XpSC33 emiRFP703-cGAS mRuby3-STING cells were transduced with shTREX1-encoding virus and analyzed via live-cell imaging after Cas9-sgF21 expression (Fig S6B). Of the four tested shTREX1 sequences (shTREX1-A, shTREX1-B, shTREX1-C, and shTREX1-D), two caused growth abnormalities. shTREX1-A, being the most effective, was chosen for further analysis (Fig S6C and D). MN formation and m-CAM events in the mCitrine-positive lineage were not affected by shTREX1-A (Fig S6E and F). These cells, however, showed no increase in sustained STING activation after m-CAM (Fig S6G and H). After treatment with reversine, an MPS1 inhibitor, for 24 h, *TREX1* knockdown led to increased STING activation (Fig S6I and J), consistent with the prior report

(Mohr et al, 2021) and confirming effective *TREX1* suppression. These results suggest that TREX1 is not the sole modulator of cGAS suppression within MN.

## STING activation after irradiation is independent of MN formation

To clarify the reasons for discrepancies between our findings and prior reports (Dou et al, 2017; Glück et al, 2017; Harding et al, 2017; Mackenzie et al, 2017), we assessed St-AI after MN formation induced by gamma-ray irradiation. XpSC33 emiRFP703-cGAS mRuby3-STING cells were transduced with a virus encoding full-length mCitrine-NLS to visualize nuclei, irradiated at 1 Gy, and subjected to live-cell imaging. As expected, irradiated cells exhibited MN as cytosolic mCitrine foci after the first mitosis post-irradiation (Figs 5A and B and S7A), which is comparable to SCF-induced MN formation (Fig 2C). Initially, we examined the cGAS localization pattern to MN and observed that only 10.3% and 9.4% of MN-positive cells exhibited the i-CAM event during the second and third interphase, respectively, whereas 77.8% and 92.3% of cells that entered mitosis displayed the m-CAM event in the second and third mitosis, respectively (Fig 5C). These results suggest that m-CAM is common in the initial MN capture by cGAS. Subsequently, we analyzed St-AI during interphase after i-CAM and m-CAM events. Among 17 i-CAM events observed, 11 cells did not show a St-AI increase after the i-CAM event (Fig S7B and C). Two cells showed a sharp St-AI increase after the i-CAM event (Fig S7D), and four did not show such a spike but sustained STING activation (Fig S7E and F). However, among the six cells that exhibited STING activation, five of them showed sustained STING activation before the i-CAM event (Fig S7D and E). This result suggests that i-CAM has a potential to trigger STING activation, but in most cases, it is not sufficient, and STING is activated by other stimuli. In agreement with this assumption, both the MN-negative lineage and the interphase after m-CAM exhibited a similar increase in St-AI (Figs 5A and D–F and S7G), suggesting that STING is activated irrespective of MN formation after 1 Gy IR exposure. Cells expressing emiRFP703-cGAS$^{R236A–R255E}$ exhibited an increased frequency of sustained STING activation in both MN-negative and MN-positive lineages (Fig 5G–I), suggesting that nucleosomal DNA leaked into the cytoplasm, which could not be visualized by mCitrine-NLS nor emiRFP703-cGAS, inhibited cGAS activation in irradiated cells.

**Figure 4. m-CAM does not lead to STING activation in subsequent interphase.**
**(A, B)** Representative images showing colocalization of mRuby3-STING with the Golgi apparatus (A) or pSTING-S366 (B) in XpSC33 emiRFP703-cGAS mRuby3-STING cells post–compound 3 treatment (1 $\mu$M, 2 h). White dot lines represent cell boundaries. Scale bar: 10 $\mu$m. **(C, D)** Percentage of pSTING-S366–positive cells (green) and cells with St-AI greater than 2.0 (magenta). Cells were treated with 1 $\mu$M compound 3 for indicated hours (C) or with indicated doses for 5 h (D) (n ≥ 35/experiment (C) or ≥ 38/experiment (D); three biological replicates). **(E)** qRT-PCR analysis of *CXCL10* mRNA levels normalized to *GAPDH* in cells treated with indicated doses of compound 3 (5-h exposure, 25-h recovery; n = 3 biological replicates). **(F)** Live-cell images of GFP and mRuby3-STING post–GFP plasmid transfection. Scale bar: 50 $\mu$m. **(G)** St-AI time-course analysis after plasmid transfection in (F). Bold lines indicate the mean. **(H)** St-AI comparison in XpSC33 emiRFP703-cGAS mRuby3-STING cells expressing shScramble or shcGAS 30 h after plasmid transfection or compound 3 treatment (n = 20). **(I, J, K)** St-AI time-course analysis after m-CAM event or plasmid transfection in XpSC33 mRuby3-STING cells expressing indicated emiRFP703-cGAS variants. In (J), cells entering mitosis with MN were analyzed in the following interphase. **(L)** Percentage of sustained STING activation in (I, J, K). **(M)** Heatmap of ISG mRNA levels normalized to *GAPDH* in XpSC33 cells. Results from total sgEMPTY cells, mCitrine- or mCerulean3-positive Cas9-sg21 cells, and cells exposed to compound 3 (1 $\mu$M, 5-h exposure, 25-h recovery). Data information: **(C, D, E)** mean ± SD, *$P < 0.05$ and ****$P < 0.0001$ (ordinary one-way ANOVA followed by Tukey's multiple comparison). **(H)** Bars represent the median, **$P < 0.01$ and ****$P < 0.0001$ (ordinary one-way ANOVA followed by Tukey's multiple comparison test). **(L)** Chi-square test.
Source data are available for this figure.

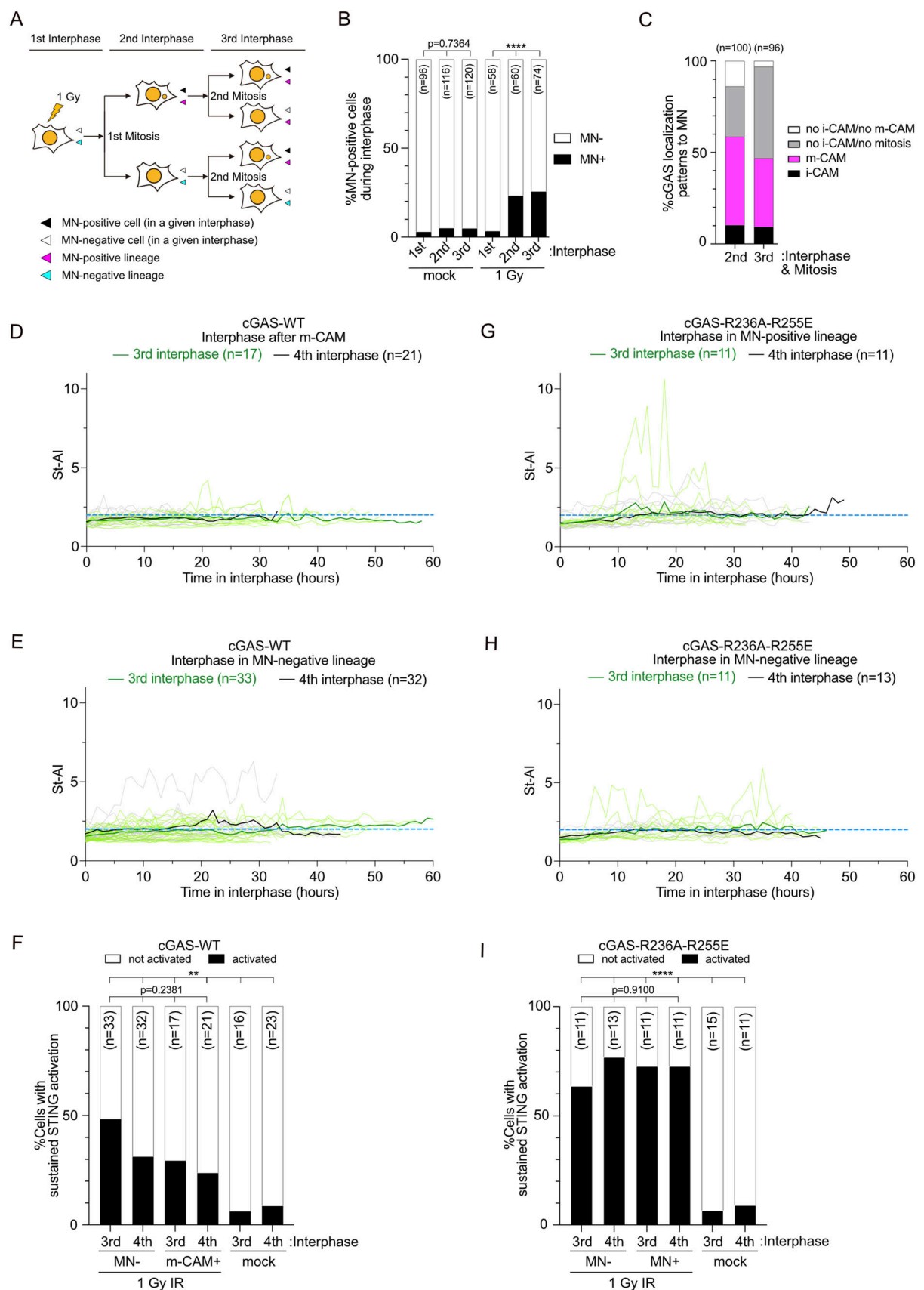

## STING activation after irradiation is linked to cytosolic mtDNA release

We asked whether STING activation after irradiation is induced by the release of mtDNA from stressed mitochondria. Cytosolic fractions from XpSC33 mCitrine-NLS emiRFP703-cGAS mRuby3-STING cells, collected 3 d after irradiation, underwent genomic qPCR with primer sets targeting mitochondrially encoded NADH dehydrogenase 1 and 2 (*MT-ND1* and *MT-ND2*). In untreated cells, both primer sets failed to amplify any products (Fig 6A). In stark contrast, cytosolic fractions from irradiated cells exhibited a substantial increase in mtDNA (Fig 6A). To mitigate this mtDNA release, we targeted VDAC1, a mitochondrial outer membrane channel protein facilitating mtDNA release into the cytosol (Guan et al, 2023). Administering 200 nM DIDS, a VDAC1 inhibitor, from 1 h pre-irradiation to 3 d post-irradiation, significantly reduced cytosolic mtDNA levels without altering total mtDNA (Fig 6A). DIDS-treated cells still activated STING upon plasmid transfection (Fig 6B and C) and showed increased MN after 1 Gy IR (Fig 6D). Notably, DIDS treatment almost entirely suppressed STING activation after IR (Fig 6E–H), suggesting that irradiation-induced STING activation is promoted by cytosolic mtDNA release. Supporting this, *TREX1* knockdown amplified STING activation in irradiated cells (Fig 6I), suggesting that TREX1 mitigates cGAS activation by digesting cytosolic mtDNA fragments.

# Discussion

In this study, we aimed to rigorously evaluate the potency of MN as an activator of the cGAS/STING pathway. Our FuVis2 reporter system allows the visualization of the nucleus in cells that have acquired a single SCF on the X chromosome, serving as an ideal reporter to assess cGAS/STING activity after MN formation without compromising mitochondrial integrity. Importantly, MN are almost exclusively derived from chromosome fusion in this reporter, which emulates MN formation in the early tumorigenesis stage called telomere crisis (Nassour et al, 2019).

We have successfully introduced cGAS and STING reporters into the FuVis2 reporter cells and confirmed that the accumulation of STING quantified as the St-AI provides a good-quality indicator of STING activation, which is validated by pSTING-S366 and downstream *CXCL10* expression. Our live-cell data suggest that chromosomes in MN can be captured by cGAS in interphase and mitosis through nuclear envelope rupturing and NEBD, respectively. In contrast to previous reports that emphasized the former i-CAM event (Harding et al, 2017; Mackenzie et al, 2017), our results suggest that the primary pathway of MN–chromatin detection by cGAS is through the latter m-CAM event, which depends on the

nucleosome-binding motif of cGAS and histone H3K79me2–mediated exposure of the cGAS-interacting acidic patch of H2A-H2B. This mechanism is distinct from cGAS localization to PN-derived chromosomes during mitosis, which may depend on DNA-binding surfaces residing in K173-I220 and H390-C405 in cGAS (Gentili et al, 2019).

Although about one third of post–m-CAM G1 cells exhibited cytoplasmic cGAS focus formation, St-AI analysis indicated that m-CAM does not lead to activation of cGAS and STING in the following interphase. Neither STING activation nor ISGs up-regulation was observed in mCitrine-positive XpSC33 Cas9-sgF21 cells, suggesting that, contrary to the previous report (Flynn et al, 2021), not only MN but also chromatin bridges caused by SCF do not activate cGAS efficiently. Moreover, neither cGAS[R236A–R255E] nor cGAS[20A] mutants could activate STING after m-CAM. It is less likely that the emiRFP703-tag abolished cGAS[R236A–R255E] enzymatic activity, because emiRFP703-cGAS[R236A–R255E]–expressing cells showed increased STING activation after irradiation. We assume that both nucleosome-binding and N-terminus hyperphosphorylation mechanisms, as well as other inhibitory mechanisms including BAF and TREX1 (Guey et al, 2020; Mohr et al, 2021), redundantly suppress cGAS activation upon MN formation, although *TREX1* knockdown alone was not sufficient to activate cGAS. We do not exclude the possibility that cGAS is slightly activated at the level that is not sufficient to induce STING activation. Although these possibilities need to be addressed in future studies, our data strongly suggest that chromatin in MN is not a potent activator of the cGAS/STING pathway and that cGAS accumulation in MN is not a reliable marker of its activation.

The idea that chromatin is inert to cGAS even in the cytosol is also supported by the absence of cGAS activation by confinement-induced PN envelope rupture (Gentili et al, 2019). Instead, our data from irradiated cells suggest that cGAS is activated independently of MN. We do not exclude the possibility that undetectable small chromatin fragments leaked into the cytosol might be the source of cGAS-activating DNA. However, the absence of the interferon response in SCF-induced FuVis2 cells, which potentially harbor acentric X chromosome fragments in the cytosol, argues against this possibility. Instead, our data and cumulative evidence support the notion that nucleic acids from disrupted mitochondria trigger the cGAS response in irradiated cells (Tigano et al, 2021; Guan et al, 2023). It is conceivable that cytosolic chromatin fragments rather inhibit cGAS activation in the presence of mtDNA.

Cytoplasmic chromatin fragments have been linked to inflammation and antitumor mechanisms because of their cGAS-accumulating potency (Dou et al, 2017; Glück et al, 2017; Harding et al, 2017; Mackenzie et al, 2017; Yang et al, 2017). A study in mouse embryonic fibroblast suggested that MN-positive cells, isolated by

**Figure 5. MN independence in STING activation after irradiation.**
**(A)** Schematic of cell cycle tracking after irradiation, defining MN-positive/negative cells and lineages in each interphase. **(B)** Percentage of MN-positive cells at different cell cycle stages post–1 Gy IR, analyzed in XpSC33 mCitrine-NLS emiRFP703-cGAS mRuby3-STING cells. **(C)** cGAS localization patterns in MN at indicated cell cycle stages post-irradiation. "No i-CAM/no m-CAM" category for cells with intact MN entering mitosis without m-CAM indicators. **(D, E)** St-AI time-course analysis post–m-CAM event (D) or in MN-negative lineages (E) at specified cell cycle stages post-irradiation. **(F)** Percentage of cells with sustained STING activation in (D, E). **(G, H)** St-AI time-course post-irradiation in XpSC33 mCitrine-NLS emiRFP703-cGAS[R236A–R255E] mRuby3-STING cells. Results from cells entering mitosis with MN (G) and MN-negative lineages (H) are shown. **(I)** Percentage of cells with sustained STING activation in (G, H). Data information: **(B, F, I)** \*\*P < 0.01 and \*\*\*\*P < 0.0001(chi-square test). Source data are available for this figure.

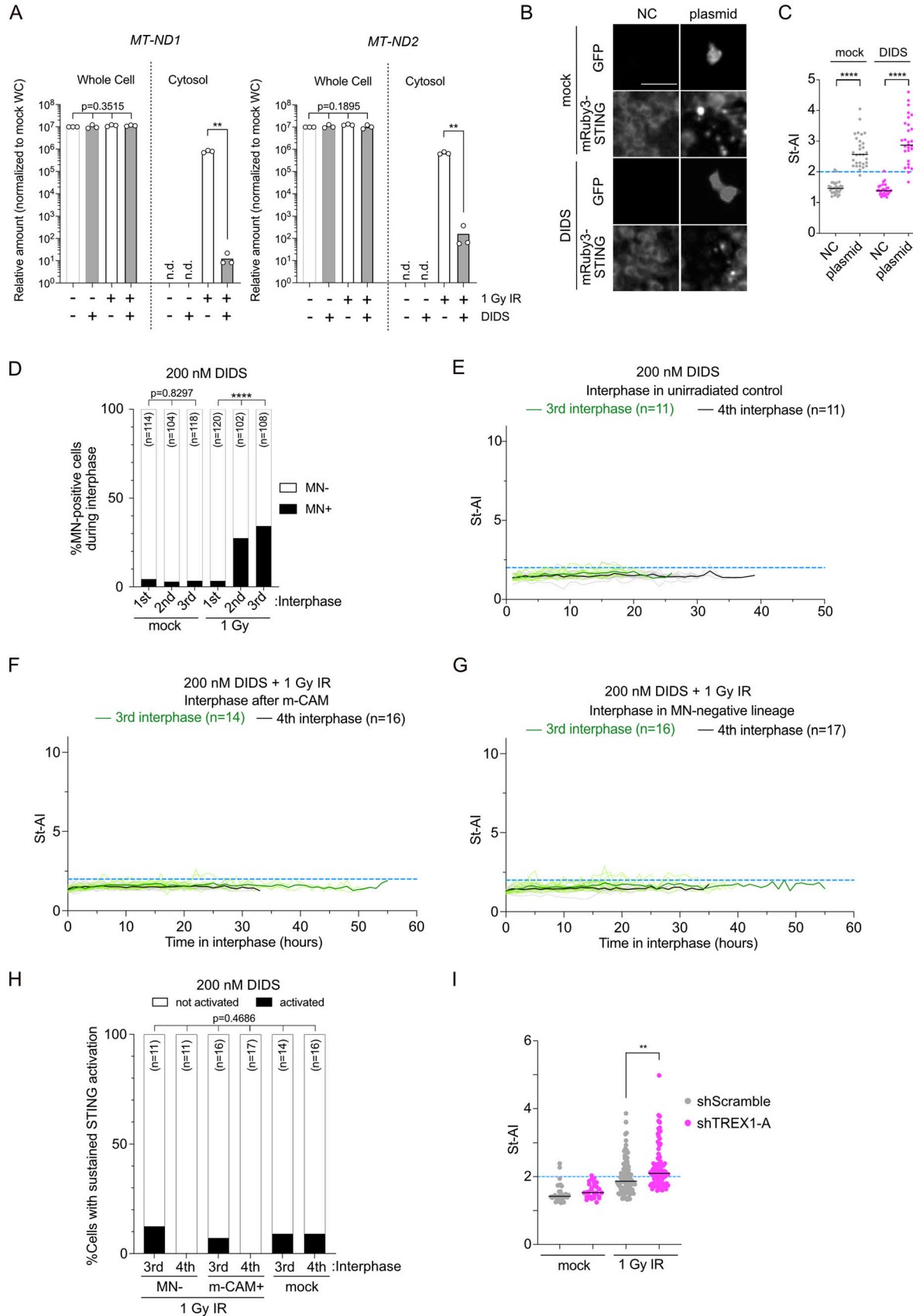

laser capture microdissection, exhibit up-regulation of ISGs (Mackenzie et al, 2017). However, recent studies have highlighted that mouse cGAS is more reactive to DNA than its human counterpart (Zhou et al, 2018) and that the overexpression of full-length mouse cGAS, but not human cGAS, activates ISGs in untreated cells (Mosallanejad et al, 2023). These studies raise the possibility that MN might activate cGAS in mice but not in human cells. Our results are in line with this hypothesis and suggest that MN are inert to a cGAS-dependent innate immune pathway in human cells. This raises further implications that MN in human cells are more prone to developing chromosome abnormalities, including chromothripsis (Zhang et al, 2015; Ly et al, 2017, 2019; Kneissig et al, 2019; Umbreit et al, 2020) and epigenetic abnormalities (Agustinus et al, 2023; MacDonald et al, 2023), even in cells with an intact cGAS/STING pathway. Although our current study is limited to a specific reporter system in a single cell line, the role of cytosolic mtDNA release in cGAS-dependent inflammatory responses in different cellular contexts with MN formation warrants careful consideration.

## Limitations of the study

Within the limitations of the FuVis2 system, it is challenging to distinguish between SCF-induced changes and other uncharacterized events that could lead to mCitrine expression. Therefore, we cannot exclude the possibility that the mCitrine-positive population includes cells that have not undergone SCF. Although this will be pursued in future studies, the potential inclusion of such uncharacterized events does not compromise our findings regarding MN and the cGAS/STING pathway. Another consideration is that our findings are based on experiments conducted in a single cell line, HCT116 cells. Given the growing evidence for species-specific variations in cGAS activity, it is possible that cGAS/STING responses may differ across various tissues and cell types within the same species. Therefore, the impact of chromosomal abnormalities, such as MN, chromatin bridges, and fragmentations, on cGAS activation in different cellular contexts remains an open question. Future investigations should particularly focus on maintaining mitochondrial integrity while assessing these effects.

# Materials and Methods

## Cell culture

Human colon carcinoma HCT116 cells (ATCC: American Type Culture Collection) and their derivatives were cultured in DMEM (Nissui

Pharmaceutical) supplemented with 10% FBS, 2 mM L-glutamine, 0.165% $NaHCO_3$, 100 U/ml penicillin/streptomycin, and 5 $\mu$g/ml Plasmocin (InvivoGen), and maintained at 37°C in 5% $CO_2$. Where indicated, the medium was supplemented with compound 3 (S8796; Selleck Chemicals) and doxycycline (Sigma-Aldrich). For DNA staining in live cells, cells were incubated with a medium containing SPY650 (Cytoskeleton) for 2 h before live-cell imaging at a one 20th concentration of the suggested concentration provided by the manufacturer's instruction. For Dot1L inhibition, a medium containing SGC0946 (S7079; Selleck Chemicals) was replaced daily for 3 d and cells were transduced with the Cas9-sgF21–encoding virus in the presence of SGC0946 up to and during live-cell imaging analysis. For Golgi staining, Cell Navigator NBD Ceramide Golgi Staining Kit (22750; AAT Bioquest) was used according to the manufacturer's instructions. For random MN induction, XpSC33 emiRFP703-Geminin iCas9-20 cells were treated with 250 nM Taxol (1097; Tocris) and 250 nM Hesperadin (24199; Cayman Chemical) for 2 d. For MPS1 inhibition, cells were exposed to 0.5 $\mu$M reversine (10004412; Cayman Chemical) for 24 h from 2 d after shTREX1-A transduction. For VDAC1 inhibition, cells were exposed to 200 nM DIDS (sc-203919A; Santa Cruz) for 1 h before irradiation. Cells were maintained in a medium containing 200 nM DIDS until harvest 3 d after irradiation.

## Plasmids

All plasmids used in this study are listed in Table S1. For cloning of the Sister-Control (SC) cassette plasmid (pMTH857) used for genomic integration, synthetic DNA fragments (Integrated DNA Technologies) were introduced into the original sister cassette plasmid (pMTH397) (Kagaya et al, 2020). A loxP sequence and two roxP sequences were inserted downstream of a 5′-exon of mCitrine/mCerulean3 and neoR-franking regions, respectively, for potential future experiments. LentiCRISPR.v2 (#52961; addgene) was mutagenized to introduce R691A to generate HiFi Cas9. pCAG-enAsCas12a-HF1(E174R/N282A/S542R/K548R)-NLS(nuc)-3xHA (#107942; addgene) was used to obtain Lenti-enAsCas12a-HF1-2C-NLS, during which one more NLS was added to the C-terminus to improve its efficiency (Liu et al, 2019). LentiGuide-puro (#52963; addgene) and an improved sgRNA scaffold sequence from pKLV2-U6gRNA5(Empty)-PGKBFP2AGFP-W (#67979; addgene) were used to generate the LentiGuide-puro-sgFUSION21-C+5 bp plasmid. pH2B-miRFP703 (#80001; addgene) and pCSII-EF-mVenus-hGeminin(1/110) (RDB15271) were used to generate pCSII-EF-emiRFP703-Geminin(1–110), during which the N-terminal sequence of miRFP703 was modified to obtain emiRFP703 (Matlashov et al, 2020). An improved rtTA3G was artificially synthesized (Integrated DNA Technologies) to obtain pLenti-rtTA3G (Zhou et al, 2006).

---

**Figure 6. mtDNA leakage leads to STING activation after irradiation.**
**(A)** qPCR analysis of total and cytosolic mtDNA in XpSC33 mCitrine-NLS emiRFP703-cGAS mRuby3-STING cells post–200 nM DIDS treatment and 1 Gy IR (n = 3 biological replicates; n.d., not detected). **(B, C)** St-AI analysis after DIDS treatment and GFP plasmid transfection. Representative images of GFP and mRuby3-STING (B) used for St-AI analysis (C) are shown (n = 30). **(D)** Percentage of MN-positive cells at different cell cycle stages after DIDS treatment and irradiation. **(E, F, G)** St-AI time-course analysis in unirradiated cells (E) and irradiated cells post–m-CAM event (F) or in MN-negative lineages (G) during DIDS treatment. **(H)** Percentage of cells with sustained STING activation in (E, F, G). **(I)** St-AI comparison in XpSC33 mCitrine-NLS emiRFP703-cGAS mRuby3-STING cells expressing shTREX1-A post-irradiation (mock, n = 30; 1 Gy IR, n = 91 and 94 for shScramble and shTREX1-A, respectively). Mock results are also shown in Fig S6J. Data information: **(A)** bars represent the mean, **P < 0.01 (ordinary one-way ANOVA for whole cell, Welch's *t* test for cytosol). **(C, I)** Bars represent the median, **P < 0.01 and ****P < 0.0001 (Welch's *t* test). **(D)** ****P < 0.0001 (chi-square test). **(H)** Fisher's exact test.
Source data are available for this figure.

pCW-Cas9 (#50661; addgene) was modified to generate pTRE3G-miRFP670nano-p2a-Cas9(HiFi), during which the puroR-t2a-rtTA sequence was removed. miRFP670nano was artificially synthesized (Integrated DNA Technologies) (Oliinyk et al, 2019). Mutagenesis on Cas9 and *CGAS* was performed by conventional PCR followed by HiFi DNA Assembly (NEB) or In-Fusion cloning (Takara Bio). Full-length sequences of plasmids used in this study are available at a public data share server (doi:10.6084/m9.figshare.24262339).

### Establishment and validation of FuVis2-XpSC cell clones

The Sister-Control (SC) reporter cassette (pMTH857) was integrated into a telomere-adjacent subtelomere sequence on the short arm of the X chromosome in HCT116 cells through CRISPR/Cas9-directed homology-mediated recombination facilitated by pMTH393, as described previously (Kagaya et al, 2020) (Fig 1A). We successfully isolated 56 independent G418-resistant clones during this process. Subsequently, we validated 10 clones (SC1, SC4, SC10, SC11, SC14, SC16, SC29, SC33, SC45, and SC53) for their intended integration using genomic PCR (Fig S1A). Quantitative PCR analysis of the integrated reporter cassette revealed that three clones (SC10, SC45, and SC53) carried two or more copies of the integrated reporter (Fig S1B). Besides these three clones, one clone (SC11) displaying an exceptionally low growth rate was excluded from the pool of candidate clones (Fig S1B). Further examination of the X chromosome structure in the remaining six candidate clones was conducted through FISH analysis using DNA probes spanning the whole X chromosome (chrX) and the X centromere (cenX). The results indicated that two clones (SC29 and SC33) harbored relatively normal X chromosomes (Fig S1B and C). Because the clone SC29 exhibited tetraploidy within the population (Fig S1D), we chose the clone SC33 for subsequent analysis.

### Establishment and validation of FuVis2-XpSC33-iCas9 cell clones

To establish XpSC33 cells featuring doxycycline (dox)-inducible HiFi SpCas9, the cells were transduced with two independent viruses carrying rtTA3G under a constitutive promoter (pMTH1190) and miRFP670nano-p2a-Cas9(HiFi) under the tight TRE promoter (pMTH1197), respectively (Fig S2A). The infected cells were treated with 1 µg/ml dox for 2 d, and miRFP670nano-positive cells were sorted using the SH800S cell sorter, which was followed by single-cell subcloning (Fig S2A). The resulting 23 subclones were subjected to a 2-d dox treatment and subsequent FACS analysis to confirm the dox-dependent miRFP670nano expression (Fig S2B). We identified five candidate subclones (iCas9-10, iCas9-13, iCas9-16, iCas9-17, and iCas9-20), which displayed more than 50% miRFP670nano-positive cells and exhibited a substantial increase of more than 1,000 times in miRFP670nano-positive cells upon dox treatment (Fig S2B). FISH analysis using chrX and cenX probes revealed that iCas9-16 harbored a translocation on the X chromosome (Fig S2C). To assess Cas9 efficiency, we transduced the candidate subclones with a virus carrying a Cas9 reporter sequence and analyzed them 4 d post-transduction (Fig S2D). This analysis revealed that iCas9-10 and iCas9-20 displayed efficient GFP targeting activities upon dox exposure, with minimal background activities (Fig S2E). Inspection of

the copy numbers of the SC reporter cassette revealed that iCas9-10 carried a duplication of the SC reporter cassette (Fig S2F). Given these assessments, we have selected an XpSC33 iCas9-20 subclone for subsequent analysis.

### Reporter cassette copy-number analysis

The Wizard Genomic DNA Purification kit (Promega) was used to extract whole-cell DNA from candidate clones, following the manufacturer's instructions. Quantitative PCR analysis (Applied Biosystems StepOnePlus Real-Time PCR) was performed using a plasmid carrying one copy of both AAVS1 and mCitrine sequences (pMTH864) as a standard, and the AAVS1 locus on the genome (two copies) as an internal control. The primers used for genomic qPCR are listed in Table S2.

### Virus transduction

Lentivirus particles were generated as previously described (Kagaya et al, 2020) with minor modifications. Briefly, 1.6 µg of a transfer plasmid was transfected into HEK293FT (for Fig 1B) or LentiX 293T cells (Clontech Laboratories, Inc.) with 0.8 µg of psPAX2 (#12260; addgene) and 0.8 µg of pCMV-VSV-G (#8454; addgene) using 9.6 µl of 1 mg/ml polyethylenimine (PEI) in a six-well plate. The medium was replaced on the next day, and the medium containing lentivirus particles was collected on days 2 and 3 post-transfection and filtered through a 0.45-µm PES syringe filter (TELS25045; Technolabs Inc.). For lentivirus infection, the medium of target cells was replaced with a virus-containing medium supplemented with 8 µg/ml polybrene. Viral titers required for near 100% transduction were empirically determined by serial dilution of the virus-containing medium, followed by antibiotic selection if applicable. We repeatedly observed that LentiX 293T cells produce higher titer lentivirus than HEK293FT cells. For the generation of cGAS and STING reporter–expressing cells, transduced cells were sorted three times by the SH800S cell sorter (Sony) with 130-µm sorting chips (Sony). For the shRNA-resistant cGAS experiment, XpSC33 cells were first transduced with shRNA-resistant cGAS mutants and selected with 10 µg/ml blasticidin from day 2. On day 3 post-transduction, some of these cells were transduced with shcGAS-encoding virus for immunoblotting on day 7, whereas others were cotransduced with shcGAS-encoding and Cas9-sgF21–encoding viruses for live-cell analysis beginning on day 7. For LentiCRISPR(HiFi) (Vakulskas et al, 2018), Lenti-enAsCas12a-HF1-2C (Kleinstiver et al, 2019), LentiGuide-sgRNA, and pLKO.1-shRNA, transduced cells were selected by 1 µg/ml puromycin for 2 d after day 2 of transduction. The following guide sequences and shRNA sequences were used (5′ to 3′): sgFusion11, GTAGCGAACGTGTCCGGCGT; sgFusion21, ATTCTACCACGGCAGTCGTT; sgFusion22, GAACGTTGGCACTACTTCAC; sgFusion23, GTGGTAGAA-TAACGTATTAC; sgFusion24, GGATCCGTAGCGAACGTGTC; sgFusion25, AACGCCGGACACGTTCGCTA; sgFusion26, CGTTCCGGTCACTCCAACGC; crFusion6, AATAATGCCAATTATTTAAA; crFusion7, AATAATTGGCATTATT-TAAA; crFusion8, AATAATGCCAATTATTTAAA; crFusion9, AGAAAAGC-GATTTGGATTA; crFusion10, GATTATAACTTCGTATAGCA; crFusion11, AAGTTAAATTCATAACTTCG; crFusion12, ACTTTAAATAATGCCAATTA; crFusion13, ACTTTAAATAATTGGCATTA; crFusion14, AAGTTAAATT-CACTCCAGA; shScramble, CCTAAGGTTAAGTCGCCCTCG; shcGAS,

TTAGTTTTAAACAATCTTTCCT; shTREX1-A, AACACGGCCCAAGGAAGAGCT (Li et al, 2017); shTREX1-B, AAGACCATCTGCTGTCACAAC (Li et al, 2017); shTREX1-C, AAGGACCCTGGAGCCCTATCC (Li et al, 2017); and shTREX1-D, CAAGGATCTTCCTCCAGTGAA (TRCN0000011206). For the generation of dox-inducible Cas9 (iCas9) cells, XpSC33 cells were simultaneously transduced with viruses encoding rtTA3G (pMTH1190) and TRE promoter–driven miRFP670nano-p2a-Cas9(HiFi) (pMTH1197), exposed to 1 $\mu$g/ml doxycycline at 2 d post-transduction for 2 d, and sorted for miRFP670nano expression by the SH800S sorter with 130-$\mu$m sorting chips. For the generation of emiRFP703-Geminin–expressing cells, XpSC33-iCas9-20 cells were transduced with lentivirus encoding emiRFP703-Geminin (pMTH1094), a derivative of the FUCCI reporter for visualization of cells in S/G2/M phases of the cell cycle (Sakaue-Sawano et al, 2008). Then, emiRFP703-positive and emiRFP703-negative cells were sequentially sorted by the SH800S sorter with 11-d intervals to enrich cells properly expressing the Geminin reporter. For the irradiation experiment, cells were transduced with lentivirus encoding mCitrine-NLS (pMTH1527) 4 d before irradiation.

## Flow cytometry

Cells were collected by trypsinization, resuspended in cold 1x PBS containing 0.1 mM EDTA, and filtered through a 5-ml polystyrene round-bottom tube with a cell-strainer cap (Corning). Cells were analyzed using the SH800S cell sorter with 100- or 130-$\mu$m sorting chips (Sony). Single cells were gated based on their low FSC-W value before analysis and sorting. Fluorescence signals were detected using the following laser and filter combinations: DAPI and BFP, 405-nm laser, 450/50 filter; mCerulean3, 488-nm laser, 450/50 filter; GFP and mCitrine, 488-nm laser, 525/50 filter; mScarlet and mRuby3, 561-nm laser, 600/60 filter; and miRFP670nano and emiRFP703, 638-nm laser, 665/30 filter.

## Gamma-ray irradiation

Two days before gamma-ray irradiation, cells were seeded onto a 35-mm dish. Subsequently, the cells were exposed to 1 Gy of $\gamma$-rays using the Cs-137 Gammacell 40 Exactor (Best Theratronics Ltd.). After irradiation, live-cell imaging was promptly carried out on the irradiated cells.

## Live-cell imaging

For the FuVis2 reporter experiment, XpSC33 and its derivative clones were transduced with lentivirus encoding Cas9-sgF21 or sgF21 (for iCas9-20 cells). Subsequently, these cells were seeded onto conventional cell culture dishes or plates at 2 d post-infection and subjected to live-cell imaging at 4 d post-infection. Live-cell imaging was performed as previously described (Kagaya et al, 2020). Briefly, cell culture dishes or plates were positioned on the BZ-X710 fluorescence microscope (KEYENCE), which was equipped with a metal halide lamp, stage-top chamber, and temperature controller featuring a built-in $CO_2$ gas mixer (INUG2-KIW; Tokai Hit). Each fluorescence signal was detected using the following filter cubes (M square): mCitrine (ex: 500/20 nm, em: 535/30 nm, dichroic: 515LP); GFP (ex: 470/40 nm, em: 525/50 nm, dichroic: 495LP); mScarlet and

mRuby3 (ex: 545/25 nm, em: 605/70 nm, dichroic: 565LP); and emiRFP703 and SPY650 (ex: 620/60 nm, em: 700/75 nm, dichroic: 660LP). Images were captured using the BZ-H3XT time-lapse module, typically at intervals of 12 or 15 min, over a duration exceeding 60 h. For the GFP plasmid control, cells were grown in a 12-well dish and transfected with pMAX-TurboGFP (pMTH380). A total of 1 $\mu$g plasmid was mixed with 5 $\mu$l of PEI (2 $\mu$g of plasmid with 10 $\mu$l of PEI for DIDS-treated cells) in 100 $\mu$l of Opti-MEM (Thermo Fisher Scientific) for 30 min before transfection. The formation of MN, the localization pattern of cGAS, and the St-AI were analyzed through manual inspection. In live-cell imaging analysis after irradiation, cells with and without mCitrine-positive MN are identified as MN-positive and MN-negative cells, respectively, in each interphase. All descendant cells originating from an MN-positive cell are defined as the MN-positive lineage. Note that a cell in the MN-positive lineage can become MN-negative in different interphases, whereas all cells in the MN-negative lineage remain MN-negative throughout.

## St-AI analysis

The cellular membrane of a target cell in the phase-contrast channel was manually inspected and tracked at 60-min intervals using the freehand selection tool within Fiji software (Schindelin et al, 2012). The tracked data were organized and stacked within the ROI (region of interest) manager. Subsequently, the stacked ROI data were superimposed onto the red channel (mRuby3-STING) to measure both maximum and mean intensities of mRuby3-STING within each cell lineage. For every cell and time-point, the maximum intensity of mRuby3-STING was divided by its mean intensity, resulting in the calculation of the STING Accumulation Index (St-AI). For GFP transfection and lentivirus infection controls, only cells that expressed fluorescent proteins were analyzed.

## Micronucleus isolation

MN isolation was performed as previously described (Mohr et al, 2021) with minor modifications. Briefly, XpSC33-iCas9-20 emiRFP703-Geminin cells were transduced with a virus encoding sgF21 and cultured in a medium containing 1 $\mu$g/ml doxycycline for 8 d. The cells were subsequently sorted based on their emiRFP703-Geminin expression using the SH800S sorter (Sony) to enrich cells in S/G2/M phases of the cell cycle. After sorting, the cells were washed and then lysed using a lysis buffer (10 mM Tris–HCl, pH 8.0, 2 mM magnesium acetate, 3 mM calcium chloride, 0.32 M sucrose, 0.1 mM, pH 8.0, EDTA, and 0.1% Nonidet P-40). Putative MN and PN fractions were subsequently collected by sucrose gradient centrifugation. This process involved mixing 10 ml of the cell lysate with 15 ml of 1.6 M sucrose buffer and 20 ml of 1.8 M sucrose buffer, both containing 5 mM magnesium acetate and 0.1 mM EDTA, pH 8.0. The centrifugation was carried out at 950$g$ for 20 min at 4°C. The obtained putative PN and MN fractions were diluted with five times their volume in cold 1x PBS and centrifuged again at 950$g$ for 20 min at 4°C. After centrifugation, supernatants were discarded, and the pellet was resuspended in cold 1x PBS/0.1 mM EDTA with 0.1 $\mu$g/ml DAPI for subsequent sorting.

## Fluorescent in situ hybridization

For mitotic chromosome spread, XpSC33 Cas9-sgF21 cells were exposed to 100 ng/ml colcemid on day 6 post-infection for 16 h to enrich mitotic cells. Subsequently, the cells were sorted based on mCitrine and mCerulean3 fluorescent cells using the SH800S sorter. The sorted cells were pelleted and then exposed to a 5 ml solution of 75 mM KCl for 7 min at room temperature. The swelling process was halted by adding 0.5 ml of ice-cold 3:1 methanol/acetic acid, and the cells were pelleted again for fixation in a 5 ml ice-cold 3:1 methanol/acetic acid solution. After centrifugation and resuspension in fresh ice-cold 3:1 methanol/acetic acid, the cells were deposited onto glass slides. After air drying, the cells were mounted with an XCP X orange probe specific for the entire X chromosome (MetaSystems Probes) and an XCE X/Y green/orange probe for X/Y chromosome centromeres (MetaSystems Probes), following the manufacturer's instructions. For samples enriched with MN and PN, sorted samples were centrifuged at 950$g$ for 20 min at 4°C to eliminate the supernatant. The pellets were then resuspended in 150 $\mu$l of ice-cold 3:1 methanol/acetic acid, and the samples were deposited onto glass slides. After air drying, the samples were mounted with the XCP X orange probe (MetaSystems Probes) beneath coverslips, heated at 75°C for 2 min, and incubated at 37°C overnight. Slides were subjected to washing with 0.4 x SSC at 72°C for 2 min and 2 x SSC with 0.05% Tween-20 at room temperature for 30 s, followed by rinsing with distilled water. After a brief drying period, samples were mounted using PNG anti-fade (4% n-propyl gallate, 100 mM Tris, pH 8.5, and 90% glycerol) with 0.1 $\mu$g/ml DAPI. Chromosome abnormalities were manually inspected with the following definition: translocation, non-X chromosome fragment on chrX; truncation, loss of chrX arm; SCF, sister chromatid fused; RING, fusion between the long- and short-arm telomeres of chrX; X/non-X, the presence of one cenX and one non-cenX centromere on a single chromosome; acentric translocation, acentric fragment of chrX translocated to another chromosome; acentric fragments, small fragments of chrX without cenX signal; and loss, no chrX/cenX signal.

## Western blotting

Typically, 10 million cells were lysed using 1 ml of 1x Laemmli sample buffer complemented with 2% 2-mercaptoethanol and 2% Benzonase (EMD Millipore). After lysis, the samples were incubated for 1 h at 37°C, followed by 10 min at 98°C. Lysates corresponding to 1.5–6.0 × 10^4 cells were separated on 4–20% Mini-PROTEAN TGX precast gels (Bio-Rad) and transferred onto PVDF membranes (Millipore). For immunoblotting, membranes designated for anti-GAPDH and anti-TREX1 were blocked for 30 min at room temperature with Blocking One (Nacalai), whereas those for anti-cGAS were blocked with 5% skim milk. The following primary antibodies were used at indicated dilution: rabbit anti-cGAS (26416-1-AP; 1:2,000; Proteintech), mouse anti-TREX1 (sc-271870; 1:1,000; Santa Cruz), and mouse anti-GAPDH (M171-3; 1:1,000; MBL). The secondary antibodies were HRP-linked anti-mouse (NA931; 1:5,000; GE Healthcare) and anti-rabbit (7074S; 1:10,000; Cell Signaling). Each membrane was cut before blocking or primary antibody application. Antibodies on the membrane were detected using

the ECL reaction and imaged with a ChemiDoc Touch imaging system (Bio-Rad). Exposure time and signal intensity were adjusted during image acquisition. No digital processing except cropping was performed on the image data. In the mScarlet-3FL-cGAS blotting, three distinct bands were detected using the anti-cGAS antibody (Fig S3A and D). The top band is likely the uncleaved blastR-p2a-mScarlet-3FL-cGAS peptide, ~105 kD in size. The middle band appears to correspond to the mScarlet-3FL-cGAS, with an estimated size of 91 kD. The bottom band is presumed to result from the cleavage of mScarlet during the maturation process, a phenomenon commonly observed in RFPs, including the mScarlet precursor, DsRed (Mizuno et al, 2003; Bindels et al, 2017).

## Immunofluorescence

Cells were cultured on coverslips coated with Alcian Blue 8GX (A5268; Sigma-Aldrich), fixed with 4% paraformaldehyde in 1x PBS for 15 min at room temperature, and washed with 1x PBS three times. The fixed cells were permeabilized using 0.2% Triton X-100, 0.02% skim milk (Nacalai), and 0.02% BSA (Sigma-Aldrich) in 1x PBS for 5 min at room temperature in dark. After rinsing with 1x PBS once and then with PBST (0.1% Tween-20, 1x PBS), the cells were incubated with the following primary antibodies diluted in PBST for 45 min at room temperature: rabbit anti-phospho-STING (Ser366) (19781S; 1:200; Cell Signaling Technology); rabbit anti-H3K79me2 (ab2594; 1:200; Abcam); rabbit anti-H3K27me3 (9733T; 1:200; Cell Signaling Technology); and mouse anti-TREX1 (sc-271870; 1:200; Santa Cruz). After three washes with PBST, the cells were incubated with Alexa Fluor 488–conjugated goat anti-rabbit (A11034; Invitrogen), Alexa Fluor 594–conjugated goat anti-rabbit (ab150080; Abcam), or Alexa Fluor 568–conjugated goat anti-mouse (A11031; Invitrogen) at a 1:1,000 dilution in PBST for 45 min at room temperature in dark, and then washed with PBST and Milli-Q water. After air drying, coverslips were mounted on glass slides using PNG anti-fade supplemented with 0.1 $\mu$g/ml DAPI. For pSTING-S366, the average intensity within a cell boundary was analyzed by ImageJ software.

## Quantitative reverse transcription PCR (qRT-PCR)

Total RNA was extracted from cells by RNeasy Mini Kit (QIAGEN). Then, 0.165 $\mu$g of total RNA was reverse-transcribed using 62.5 nM Oligo dT and 0.18 $\mu$l of AMV reverse transcriptase (NIPPON GENE) in a total 25 $\mu$l reaction mix following the manufacturer's instructions. The resulting cDNA was used for qPCR with THUNDERBIRD Next SYBR qPCR Mix (Toyobo) and StepOnePlus Real-Time PCR System (Applied Biosystems). For ISGs, mean values of three biological replicates were visualized as a heatmap using the online tool Heatmapper (Babicki et al, 2016). The primers used are indicated in Table S2.

## Cytosolic fraction and whole-cell DNA isolation for mtDNA quantification

The isolation of the cytosolic fraction was conducted with minor modifications to a previously established protocol (Guan et al, 2023). One million cells were lysed in 100 $\mu$l of digitonin buffer

(150 mM NaCl, 50 mM Hepes, pH 7.4, and 25 µg/ml digitonin) and incubated on a rotator for 10 min at 4°C. The lysed sample was centrifuged at 2,000$g$ for 10 min, and the supernatant was transferred to a new tube. Subsequently, the supernatant was subjected to centrifugation again at 2,000$g$ for 20 min, and the resulting supernatant was once more transferred to a new tube. This centrifugation and transfer process was repeated three additional times, and the final supernatant obtained was used for qPCR analysis. The primers used are indicated in Table S2. Whole-cell DNA was extracted from one million cells using the Wizard Genomic DNA Purification kit (Promega), following the manufacturer's instructions, and resuspended in 100 µl of DNA Rehydration Solution.

### Statistical analysis

All statistical analyses and graphing were performed using GraphPad Prism software (version 10.0). The specific statistical tests applied to each dataset are detailed in the corresponding figure legends. We established a significance threshold (alpha level) at 0.05 for all analyses.

# Data Availability

All data are archived at Kyoto University and available from the corresponding author upon reasonable request. Full-length DNA sequences of plasmids used in this study are available at a public data share server (doi:10.6084/m9.figshare.24262339).

# Supplementary Information

# Acknowledgements

We thank the Drug Discovery Centre, supported by the iSAL (Innovative Support Alliance for Life Science), Kyoto University, for the cell sorter; CORE Program of the Radiation Biology Center, Kyoto University, for gamma-ray irradiation; the RIKEN BRC through the National BioResource Project of the MEXT, Japan, for material distribution; Dorus Gadella, Fuyuki Ishikawa, Keith Joung, Masato Kanemaki, Jan Karlseder, Minsoo Kim, Benjamin Kleinstiver, Eric Lander, Atsushi Miyawaki, David Sabatini, Ryota Sato, Tomohiko Taguchi, Didier Trono, Vladislav Verkhusha, Robert Weinberg, Kousuke Yusa, and Feng Zhang for sharing materials; Yumi Hayashi for assistance with molecular cloning and St-AI analysis; Andrea Ruelas-Gonzalez and Yuya Nishida for assistance with St-AI analysis; and members of the Hayashi laboratory for their suggestion and discussion. This project was supported by grants from the Grant-in-Aid for Scientific Research (B) (20H03183) to MT Hayashi; and institutional funding from the Hakubi Center and the Graduate School of Medicine, Kyoto University, to MT Hayashi.

### Author Contributions

Y Sato: conceptualization, formal analysis, validation, investigation, visualization, methodology, and writing—original draft.

MT Hayashi: conceptualization, resources, formal analysis, supervision, funding acquisition, validation, investigation, visualization, methodology, project administration, and writing—original draft, review, and editing.

### Conflict of Interest Statement

The authors declare that they have no conflict of interest.

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
