## [Reviewer comments · Life Science Alliance]

Life Science Alliance

Micronucleus Is Not a Potent Inducer of cGAS-STING Pathway

Yuki Sato and Makoto Hayashi

DOI: <https://doi.org/10.26508/lsa.202302424>

Corresponding author(s): Makoto Hayashi, Kyoto University

Review Timeline:

Submission Date:	2023-10-07
Editorial Decision:	2023-11-09
Revision Received:	2023-12-29
Editorial Decision:	2024-01-19
Revision Received:	2024-01-22
Accepted:	2024-01-22

Transaction Report:

November 9, 2023

Re: Life Science Alliance manuscript #LSA-2023-02424

Dr. Makoto T Hayashi
Kyoto University
The Graduate School of Medicine
Yoshida-Konoe-cho
Sakyo-ku
Kyoto, Kyoto 60608501
Japan

Dear Dr. Hayashi,

Thank you for submitting your manuscript entitled "Micronucleus Is Not a Potent Inducer of cGAS-STING Pathway" to Life Science Alliance. The manuscript was assessed by expert reviewers, whose comments are appended to this letter. We invite you to submit a revised manuscript addressing the Reviewer comments.

Thank you for this interesting contribution to Life Science Alliance. We are looking forward to receiving your revised manuscript.

Sincerely,

B. MANUSCRIPT ORGANIZATION AND FORMATTING:

Reviewer #1 (Comments to the Authors (Required)):

Comments to authors.

In this article Sato and Hayashi are reporting results that indicate that micronuclei are not a potent activator of the cGAS-STING pathway and of Type I interferon response. The experimental design is very elegant, with authors taking on the challenge to monitor in a single-cell basis chromosomal fusions, micronuclei formation and subsequent activation, or not, of cGAS DNA sensing and interferon response mediated by STING. This biological issue is important as major publications in the recent years were proposing that micronuclei activate cGAS in response to oncogenic stress, or irradiation-induced DNA damage. cGAS activation by micronuclei has been proposed to promote cellular senescence and to impact on cancer cell rejection by the immune system (Immune cell death). However, the direct evidence for cGAS positive micronuclei being able to activate cGAS-STING was still lacking.

The data are all strong and convincing. Furthermore, because the importance of the cGAS-STING pathway in aging and cancer biology, I think these results should be published for a matter of scientific debate.

I here below list a few minor issues and comments that I hope will help the authors to improve the paper before its publication.

Minor issues

1. The data presented in the paper strongly suggest that MNs are not potent activators of cGAS-STING. To this regard cytosolic DNA staining, with antibodies recognizing ds- or ss-DNA or DNA dyes (picogreen has been published in several papers) should be performed, especially after irradiation. One difference between SCF and IR, is that IR could release more potently DNA pieces compared to a single-chromosomal event.
2. According to my knowledge (and experience) it is very difficult to assess interferon induction or cGAS activation in 'physiological' conditions, let say under low/mild DNA damage or chromosomal instability (IR is a more potent DNA damage inducer and thus cGAS inducer). However, senescence phenotypes (proliferation arrest, SASP, etc...) are known to depend on the presence of cGAS. Have authors considered to assess if MN+ cells do acquire senescence features (maybe at latter time points as 10-14 days)?
3. What is very well-described in the discussion is that cells possess a line of control-mechanisms to block cGAS activation when self-DNA is released from the primary nucleus. One of this main mechanism is DNA degradation by the cytoplasmic TREX1 exonuclease. What is the level of expression of TREX1 in HCT116 cells and the clones derived in this study? It would be informative to see whether TREX1 relocate to MN after SCF. In addition, authors could deplete TREX1 in order to assess if then cGAS-STING activation can be observed in MN+ cells after SCF events. I expect that TREX1 depletion would lead to a more potent cGAS and interferon response, and would furthermore maybe promote the establishment of senescence. Compared to previous publications, mostly done in human fibroblasts and MEFs, are the HCT116 having a higher level of TREX1 that could explain their apparent resistance to MN formation?
4. P4 lines 70-71. I think it is not completely accurate to say that previous studies were only based on cell population studies. I can think of the paper of MacKenzie et al 2017 (cited by authors) that used laser microdissection to isolate cells having or not MNs, and then to perform single-cell RNA seq. MacKenzie et al. could show that after IR, MN+ cells express some SASP/ISGs factors such as ISG15. In the present study authors mostly rely on cxcl10 expression, maybe the assessment of STING induction of interferon and ISGs would be better achieved with a larger panel of genes or by RNA-seq, especially if MN+ and MN- can be sorted. Do authors think that cGAMP concentration would be a better readout of cGAS activation and would it be feasible in their experimental conditions (after sorting of cells distinguishing SCF+ to other cells)?
5. Abstract. P2 line 29. 'suggest that cGAS accumulation in the cytosol is not a robust indicator of its activation'. Should be rephrased because authors mainly report that cGAS accumulation inside micronuclei is not sufficient to activate cGAS-STING.

6. P19 line 349. Authors used the word 'nucleotides', which is incorrect and should be replaced by: nucleic acids, because cGAS is a dsDNA sensor with a minimum size recognition of around 40 bps. cGAS can also be activated by RNA:DNA hybrids.

7. P11 line 191, and Figure 3A. The level of expression of mScarlet-cGAS is not shown. Is it an overexpression? Is it present in all cells?

8. Fig4E. cxcl10 was misspelled (clcx10).

Reviewer #2 (Comments to the Authors (Required)):

Sato and Hayashi develop a reporter cell line to evaluate the consequences of micronucleation in live cells, where they aimed to improve the design of a reporter system they previously generated (Kayagi et al, 2020). The new reporter takes advantage of the identical N-terminal sequences of mCitrine and mCerulean3 and results in the expression of either fluorophore depending on the outcome of CRISPR-Cas9 cutting at sites flanking a neo resistance marker. mCitrine expression is caused by sister chromatid fusion upon CRISPR-Cas9 cutting, whereas mCerulean3 expression results from deletion of the neo resistance gene. The authors then use the reporter to investigate whether micronuclei from individual cells activate the innate immune response. They assess cellular localization of the cytoplasmic DNA sensor, cGAS, throughout the cell cycle as well as evaluating whether micronuclei lead to cGAS activation. Interestingly, they find that cGAS infrequently localizes to micronuclei in interphase. cGAS primarily binds to micronuclear DNA in mitosis and remains bound during the subsequent interphase. The authors also find that cGAS binding to micronuclear DNA does not promote innate immune response activation, supporting existing evidence that micronuclei do not activate cGAS-STING signalling.

Overall, the manuscript provides a useful resource for the community to study the consequences and fates of micronuclei in live cells over time. The study also provides interesting new insights into the cell cycle-regulated localization of cGAS to micronuclei and begins to explore the requirements for mitotic cGAS binding to micronuclear DNA. However, the idea that micronuclei do not efficiently induce innate immune proinflammatory signalling has been previously proposed by Tim Mitchison's group (Flynn et al, 2021; PMID: 34819364). Additionally, despite the utility of the resource, there are some uncertainties regarding the reporter that should be addressed as well as experimental controls that should be added to the manuscript to support the claims made by the authors before publication. Altogether, we feel that this manuscript could be a good candidate for LSA if the points below can be addressed.

Specific comments

- The authors present the FuVis2-XpSC reporter system to have one of three independent outcomes: i) neoR deletion resulting in mCerulean3 expression, ii) indel generation, or iii) sister chromatid fusion resulting in mCitrine expression. However, analysis of X chromosome abnormalities in mCitrine-sorted cells in Fig. 1D shows that only ~20% of metaphase spreads have sister chromatid fusion events. This indicates that sister chromatid fusion is likely not the only event resulting in mCitrine expression since ~100% of cells would have sister chromatid fusion events then. Given the importance of defining how the reporter system works, the authors should determine what other genomic events lead to mCitrine expression. The authors should amplify and sequence genomic sites containing the mCitrine reporter in mCitrine-sorted cells to determine what other events result in fluorescence. The authors should sequence both upstream and downstream of the integrated reporter as well to determine whether these events are occurring in the X chromosome. The authors should also modify their schematic in Fig. 1A and the text accordingly to account for this.
- Although it is true that sister chromatid fusion can generate micronuclei, this is not the only outcome-sister chromatid fusion will also generate chromosome bridges that may not be visible with the DNA detection methods used here. Moreover, Flynn et al. (PMID: 34819364) suggested that chromosome bridges not micronuclei are the true activators of cGAS after induced chromosome missegregation. Although we do not insist that this issue needs to be addressed experimentally, this caveat needs to be discussed in a 'Limitations of the study' section at the end of the Discussion. This section also needs to discuss the fact that this study only uses one cell line for its experiments.
- There are some discrepancies in data presented that should be clarified or fixed. The experiments in Fig. 1B provide different results than those in Fig. S1H, where the same cell line and methods were used to generate the data. For example, in Fig. 1B, a maximum of 6% mCerulean3-positive cells were observed 9 days post-transduction of Cas9-sgF21 whereas ~20% were observed at day 9 in Fig. S1H. Is there a reason for the large variation in fluorescent cells? Similarly, the percentage of abnormal X chromosomes in the FuVis2.0 SC29 clone is listed as 6.7% in Fig. S1B but is shown as an average of >10% in Fig. S1C. Are the values listed in Fig. S1B representative of a single replicate? If so, it should be clarified. The authors also mention in lines 145-146 that neither mCitrine- or mCerulean3-positive cells had micronuclei in the first interphase but then indicate in lines 163-164 that both mCitrine- and mCerulean3-positive populations had micronuclei in the first interphase. The plot in Fig. 2C demonstrates that there are micronuclei present during the first interphase in both cell types, agreeing with the statement in lines 163-164. Finally, in line 207, it is stated that cGAS accumulates in ~50% of MN in the subsequent interphase but the bar plot in Fig. S3D indicates that cGAS accumulates in ~30%. These issues should be clarified or adjusted accordingly in the text.

- Although the effect of the nucleosome binding mutant of cGAS on micronuclear DNA localization is clear in Fig. 3C, it is hard to interpret the negative results from the cGAS phospho-mutants since these experiments were performed in cells expressing endogenous cGAS. The experiments should be repeated following cGAS knockdown or knockout to assess the impact of the mutants on cGAS localization during mitosis. The expression of the cGAS mutants should also be evaluated by immunoblot analysis to compare to endogenous cGAS protein levels.
- Given the interesting observation that cGAS primarily binds to micronuclear chromosomes in mitosis, it would be a nice way to visualize the findings by extending the bars in Fig. 3A to include cGAS localization upon nuclear envelope breakdown (i.e., extending past timepoint '0'). It is also not too clear why the 'end of the imaging' category is included in the analyses in Fig. 3A. It would help to clarify the rationale for it.
- The authors should provide confirmation of the degree of H3K79me2 loss following seven days of DOT1L inhibitor treatment for the experiments performed in Fig. 3F. Although H3K79me2 reduction by DOT1L inhibitor has been confirmed in PMID: 36732527, those experiments were not performed in HCT116 cells and a direct comparison to interpret the experimental results is needed.

Additional points

- A parental HCT116 control sample is missing from Fig. S1D.
- Line 222: H3K79me2 is not a MN-specific histone modification. It is also found in the primary nucleus at sites of active transcription. The text should be adjusted accordingly.
- Line 358: The authors should also cite PMID: 37286600 for epigenetic abnormalities in micronuclei.
- The order of the panels in some figures can be confusing since they aren't presented in alphabetical order. For example, Fig. S3 has panel D at the top right rather than after panel C. Similar is found in Fig. 3, Fig. S1, and Fig. S4. It would help to re-organize the figure panels so that they appear in the order they are referred to in the text.

There are some grammatical errors/typos that should be addressed.

- 'Micronucleus' and 'micronuclei' are used interchangeably throughout the manuscript as the abbreviation 'MN' that was initially defined as 'micronucleus'. 'MNs' is also used as the plural of MN, which should be micronuclei not micronucleus. It might be worth changing the MN abbreviation to 'micronuclei' throughout the text.
- The abbreviation 'PN' is introduced twice in the manuscript in lines 35 and 174. The second instance should be removed and can just be referred to as PN.
- Lines 50-51: 'which phosphorylates TBK1 itself' could be re-worded to 'which then phosphorylates itself'.
- Line 51: 'the' should be added before 'interferon regulatory factor 3'.
- Lines 65 and 103: 'N-terminus' should be 'N-terminal'.
- Line 70: 'different' can be removed.
- Lines 72-73: 'This raises questions about how cGAS can be efficiently activated by MN in the presence of suppressive chromatin-cGAS interaction, with some studies suggesting that....' should be rephrased to 'This raises the question of how cGAS can be efficiently activated by MN in the presence of suppressive chromatin-cGAS interaction, with a recent study suggesting that....' since a single question and reference were provided.
- Line 82: 'have' should be removed.
- Line 84: 'has' should be removed.
- Line 87: 'opportunity' can be re-worded to 'system'.
- Line 129: 'chromosome' should be removed.
- Line 146: 'fluorescent-positive' can be re-worded to 'fluorescent'.
- Lines 156-167: 'analyzed from day 2 to 6 using a flow cytometer confirming...' should be rephrased to 'analyzed from day 2 to 6 using flow cytometry confirming...'
- Line 165: 'did' should be re-worded to 'do'.
- Line 170: 'for' should be added before 'cells'.
- Line 213: 'the' should be added before 'cGASR236A-R255E mutant'.
- Line 217: 'phosphor' should be 'phospho'.
- Line 219: 'This result indicates that....' can be re-worded to 'These results indicate that...'
- Line 226: nucleosome should be plural.
- The header in line 228 should be 'm-CAM does not lead to STING activation'.
- Lines 247, 273, 315, 326, and 329: 'cxcl10' should be capitalized and italicized.

We thank the editor and reviewers for handling our manuscript. The comments provided by each reviewer were not only critical but also constructively shaped the course of our revisions. We deeply appreciate their efforts in enhancing the quality and clarity of our work.

To facilitate easy identification of revisions, we have highlighted changes made in response to **Reviewer 1** and **Reviewer 2**'s comments in distinct colors. We have also corrected minor grammatical errors and typos throughout the manuscript, though these may not be specifically highlighted.

Key updates include the creation of Figure 6, which elucidates the role of cytosolic mtDNA in cGAS activation following irradiation. Furthermore, we have incorporated Figure S6 to explore the potential suppressive function of TREX1 exonuclease in regulating cGAS activity upon MN formation. Additionally, we have substantially revised figure legends to meet the standard of the journal guideline.

We are confident that these revisions substantially enrich the manuscript and address the insightful feedback provided by the reviewers. We look forward to the manuscript moving closer to publication.

Reviewer #1 (Comments to the Authors (Required)):

Comments to authors.

In this article Sato and Hayashi are reporting results that indicate that micronuclei are not a potent activator of the cGAS-STING pathway and of Type I interferon response. The experimental design is very elegant, with authors taking on the challenge to monitor in a single-cell basis chromosomal fusions, micronuclei formation and subsequent activation, or not, of cGAS DNA sensing and interferon response mediated by STING. This biological issue is important as major publications in the recent years were proposing that micronuclei activate cGAS in response to oncogenic stress, or irradiation-induced DNA damage. cGAS activation by micronuclei has been proposed to promote cellular senescence and to impact on cancer cell rejection by the immune system (Immune cell death). However, the direct evidence for cGAS positive micronuclei being able to activate cGAS-STING was still lacking.

The data are all strong and convincing. Furthermore, because the importance of

the cGAS-STING pathway in aging and cancer biology, I think these results should be published for a matter of scientific debate.

I here below list a few minor issues and comments that I hope will help the authors to improve the paper before its publication.

Minor issues

1. The data presented in the paper strongly suggest that MNs are not potent activators of cGAS-STING. To this regard cytosolic DNA staining, with antibodies recognizing ds- or ss-DNA or DNA dyes (picogreen has been published in several papers) should be performed, especially after irradiation. One difference between SCF and IR, is that IR could release more potently DNA pieces compared to a single-chromosomal event.

Response & Change:

We appreciate the positive and constructive comments by the reviewer. While PicoGreen is known to stain cytoplasmic mitochondrial DNA within intact mitochondria (<https://doi.org/10.1016/j.yexcr.2004.10.013>), this could potentially complicate result interpretation due to our microscope's resolution limitations in differentiating cytosolic and mitochondrial DNA. To circumvent this, we performed cytosolic fractionation followed by qPCR for mitochondrial DNA (mtDNA). Our results showed a significant increase in mtDNA in the cytosolic fraction post-irradiation (Fig. 6A).

To further investigate whether mtDNA leakage triggers STING activation, we used the VDAC1 inhibitor DIDS, known to block mtDNA release upon irradiation (doi: 10.3390/ijms24044020). We found that 200 nM DIDS treatment effectively suppressed mtDNA leakage into the cytosol (Fig. 6A) and substantially inhibited STING activation after irradiation (Fig. 6E-H). We confirmed that DIDS treatment did not impede STING activation after plasmid transfection (Fig. 6B) nor affect MN formation after irradiation (Fig. 6D). These results support the hypothesis that cytosolic mtDNA is a potent trigger of cGAS activity in the context of irradiation. We have detailed these new results in lines 328-344 and in Fig. 6.

2. According to my knowledge (and experience) it is very difficult to assess interferon induction or cGAS activation in 'physiological' conditions, let say under low/mild DNA damage or chromosomal instability (IR is a more potent DNA damage inducer and thus cGAS inducer). However, senescence phenotypes (proliferation arrest, SASP, etc...) are known to depend on the presence of cGAS. Have authors considered to assess if MN+ cells do acquire senescence features (maybe at latter time points as 10-14 days)?

Response & Change:

To address the reviewer's point, we sorted mCitrine- and mCerulean3-positive cells from the XpSC33 Cas9-sgF21 population seven days post-infection. These sorted cells were then re-cultured and examined for the expression of senescence markers, p21 and LaminB1, at 17 days post-infection. As a control, a subset of cells was treated with Bleomycin during the last three days of this period (Fig. S5B). The result suggested that mCitrine-positive cells exhibited no signs of senescence, unlike those treated with Bleomycin (Fig. S5C). This result further supports our hypothesis that MN derived from SCF does not lead to cGAS activation in this cellular context. These results have been described in lines 277-282.

3. What is very well-described in the discussion is that cells possess a line of control-mechanisms to block cGAS activation when self-DNA is released from the primary nucleus. One of this main mechanism is DNA degradation by the cytoplasmic TREX1 exonuclease. What is the level of expression of TREX1 in HCT116 cells and the clones derived in this study? It would be informative to see whether TREX1 relocate to MN after SCF. In addition, authors could deplete TREX1 in order to assess if then cGAS-STING activation can be observed in MN+ cells after SCF events. I expect that TREX1 depletion would lead to a more potent cGAS and interferon response, and would furthermore maybe promote the establishment of senescence. Compared to previous publications, mostly done in human fibroblasts and MEFs, are the HCT116 having a higher level of TREX1 that could explain their apparent resistance to MN formation?

Response & Change:

To address this point, we analyzed TREX1 localization to SCF-derived MN by

immunofluorescence and found TREX1 presence as cytosolic foci (Fig. S6A). We then evaluated four shTREX1 sequences, selecting the most efficient for further analysis (Fig. S6B-D). This process revealed that HCT116 cells express TREX1 at levels comparable to those in fibrosarcoma HT1080 cells (Fig. S6D). We confirmed that shTREX1 expressing cells exhibited increased STING activation following Reversine treatment compared to the shscramble control (Fig. S6I-J), as shown previously (doi: 10.1016/j.molcel.2020.12.037). However, enhanced STING activity was not detected following SCF-induced MN formation (Fig. S6E-H). These results suggest that TREX1 is not the sole factor suppressing cGAS activation in HCT116 cells.

While the mechanism of cGAS suppression remains an intriguing topic, we propose that this aspect may be beyond the current scope of our paper. We have included these new findings in lines 284-297 and Fig. S6 and hope the reviewer agrees with our focus.

4. P4 lines 70-71. I think it is not completely accurate to say that previous studies were only based on cell population studies. I can think of the paper of MacKenzie et al 2017 (cited by authors) that used laser microdissection to isolate cells having or not MNs, and then to perform single-cell RNA seq. MacKenzie et al. could show that after IR, MN+ cells express some SASP/ISGs factors such as ISG15. In the present study authors mostly rely on cxcl10 expression, maybe the assessment of STING induction of interferon and ISGs would be better achieved with a larger panel of genes or by RNA-seq, especially if MN+ and MN- can be sorted. Do authors think that cGAMP concentration would be a better readout of cGAS activation and would it be feasible in their experimental conditions (after sorting of cells distinguishing SCF+ to other cells)?

Response & Change:

We thank the reviewer for raising this point. A recent study indicates that mouse cGAS is more readily activated by self-DNA compared to human cGAS (doi: 10.1126/sciimmunol.abp9765; doi: 10.1016/j.cell.2018.06.026). This is relevant as MacKenzie et al. used MEFs for their single-cell RNA seq. This species-specific difference in cGAS activation could explain the discrepancies noted. We have amended the discussion to include this point (lines 396-402).

Due to the challenge in isolating purely MN-positive cells, we used mCitrine-positive XpSC33 Cas9-sgF21 cells for expanding our qPCR analysis to include a broader range of ISGs. The revised results still support our initial conclusion that STING is not activated in mCitrine-positive cells, including those with MN (Fig. 4M and S5A).

We agree that cGAS might be activated at levels insufficient to induce a STING response. We have included this in the discussion (lines 379-380). Nonetheless, we believe such possibility does not alter our conclusion that micronucleus is not a potent inducer of cGAS-STING pathway. Regarding cGAMP concentration analysis, we were cautious about potential mitochondrial integrity compromise during sampling, which could transiently activate cGAS even in postlysis samples, as recently reported (DOI: 10.1126/sciimmunol.abp9765). Therefore, we believe monitoring STING activity in living cells and analyzing downstream ISG upregulation provide more accurate reflections of cGAS activation than measuring cGAMP concentration in this context.

Responding to reviewer 2's comments, we have added a "limitations of the study" section at the end of the discussion. Here we clarify that our conclusions are based on data from a single cell line, and that the question of MN's inertness to cGAS in other cellular contexts remains to be experimentally addressed (lines 418-424).

5. Abstract. P2 line 29. 'suggest that cGAS accumulation in the cytosol is not a robust indicator of its activation'. Should be rephrased because authors mainly report that cGAS accumulation inside micronuclei is not sufficient to activate cGAS-STING.

Response & Change:

We have rephrased the sentence as "suggest that cGAS accumulation in cytosolic MN is not a robust indicator of its activation".

6. P19 line 349. Authors used the word 'nucleotides', which is incorrect and should be replaced by: nucleic acids, because cGAS is a dsDNA sensor with a minimum size recognition of around 40 bps. cGAS can also be activated by RNA:DNA hybrids.

Response & Change:

We have amended the text as suggested (line 391).

7. P11 line 191, and Figure 3A. The level of expression of mScarlet-cGAS is not shown. Is it an overexpression? Is it present in all cells?

Response & Change:

We have sorted mScarlet-positive cells three times to enrich mScarlet-cGAS expressing cells (detailed in the method section). This procedure ensured the presence of mScarlet-cGAS in all cells. To assess the expression levels of mScarlet-cGAS, we conducted immunoblotting on both parental and mScarlet-cGAS cells. The results suggest that mScarlet-cGAS is over-expressed compared to the endogenous cGAS levels (lines 191-192, Fig. S3A).

8. Fig4E. cxcl10 was misspelled (clcx10).

Response & Change:

We have revised the text.

Reviewer 2 (Comments to the Authors (Required)):

Sato and Hayashi develop a reporter cell line to evaluate the consequences of micronucleation in live cells, where they aimed to improve the design of a reporter system they previously generated (Kayagi et al, 2020). The new reporter takes advantage of the identical N-terminal sequences of mCitrine and mCerulean3 and results in the expression of either fluorophore depending on the outcome of CRISPR-Cas9 cutting at sites flanking a neo resistance marker. mCitrine expression is caused by sister chromatid fusion upon CRISPR-Cas9 cutting, whereas mCerulean3 expression results from deletion of the neo resistance gene. The authors then use the reporter to investigate whether micronuclei from individual cells activate the innate immune response. They assess cellular localization of the cytoplasmic DNA sensor, cGAS, throughout the

cell cycle as well as evaluating whether micronuclei lead to cGAS activation. Interestingly, they find that cGAS infrequently localizes to micronuclei in interphase. cGAS primarily binds to micronuclear DNA in mitosis and remains bound during the subsequent interphase. The authors also find that cGAS binding to micronuclear DNA does not promote innate immune response activation, supporting existing evidence that micronuclei do not activate cGAS-STING signalling.

Overall, the manuscript provides a useful resource for the community to study the consequences and fates of micronuclei in live cells over time. The study also provides interesting new insights into the cell cycle-regulated localization of cGAS to micronuclei and begins to explore the requirements for mitotic cGAS binding to micronuclear DNA. However, the idea that micronuclei do not efficiently induce innate immune proinflammatory signalling has been previously proposed by Tim Mitchison's group (Flynn et al, 2021; PMID: 34819364). Additionally, despite the utility of the resource, there are some uncertainties regarding the reporter that should be addressed as well as experimental controls that should be added to the manuscript to support the claims made by the authors before publication. Altogether, we feel that this manuscript could be a good candidate for LSA if the points below can be addressed.

Specific comments

- The authors present the FuVis2-XpSC reporter system to have one of three independent outcomes: i) neoR deletion resulting in mCerulean3 expression, ii) indel generation, or iii) sister chromatid fusion resulting in mCitrine expression. However, analysis of X chromosome abnormalities in mCitrine-sorted cells in Fig. 1D shows that only ~20% of metaphase spreads have sister chromatid fusion events. This indicates that sister chromatid fusion is likely not the only event resulting in mCitrine expression since ~100% of cells would have sister chromatid fusion events then. Given the importance of defining how the reporter system works, the authors should determine what other genomic events lead to mCitrine expression. The authors should amplify and sequence genomic sites containing the mCitrine reporter in mCitrine-sorted cells to determine what other events result in fluorescence. The authors should sequence both upstream and downstream of the integrated reporter as well to determine whether these events

are occurring in the X chromosome. The authors should also modify their schematic in Fig. 1A and the text accordingly to account for this.

Response & Change:

We thank the reviewer for the insightful comment. Our interpretation differs slightly, based on the nature of SCF. Since SCF is typically observed only in the first mitosis after its formation and is resolved (broken) during mitotic exit, identifying SCF in 100% of mitotic spreads in mCitrine-positive cells, which undergo multiple mitoses after SCF formation, is challenging. This interpretation aligns with observed X chromosome fragmentation in mCitrine-positive cells, potentially a result of MN formation and subsequent mitoses, as reported by David Pellman's group (doi: 10.1038/nature10802).

With this interpretation, we have attempted to capture the first mitosis upon SCF formation by a two-step approach. Initially, we determined the onset of mCitrine expression and subsequent onset of mitosis in XpSC33 Cas9-sgF21 cells (Figure below). We found mCitrine expressing cells started entering mitosis around 87 hours post-infection. Then, to enrich mCitrine-positive cells in their first mitosis, we applied colcemid to XpSC33 Cas9-sgF21 cells at 72 hours post-infection for a duration of 24 hours (Schematic below). Unfortunately, the sorting duration required (over 10 hours) was impractical for obtaining sufficient cells for chromosome spread analysis. Attempts to sort mCitrine-positive cells for 10 hours and subsequent processing for chromosome spread did not yield discernible chromosome spreads for SCF analysis. Additionally, the slower sorting process significantly increased the likelihood of non-fluorescent cell contamination.

While we acknowledge the importance of characterizing chromosome structures in mCitrine-positive cells, differentiating potential uncharacterized events leading to mCitrine expression from SCF-induced subsequent changes remains challenging. We have updated the manuscript to account for these potential uncharacterized events that lead to mCitrine expression (lines 128-132) and have discussed this limitation in a newly made "limitations of the study" section more extensively (lines 412-418). We hope the reviewer agrees that the nature of the first event does not affect our paper's overall conclusion. Detailed investigation into the structural abnormalities induced by the FuVis2 system will be pursued in our future studies.

[Figure removed by editorial staff per authors' request].

Schematic for the enrichment of mCitrine-positive cells in their first mitosis.

• Although it is true that sister chromatid fusion can generate micronuclei, this is not the only outcome-sister chromatid fusion will also generate chromosome bridges that may not be visible with the DNA detection methods used here.

Moreover, Flynn et al. (PMID: 34819364) suggested that chromosome bridges not micronuclei are the true activators of cGAS after induced chromosome missegregation. Although we do not insist that this issue needs to be addressed experimentally, this caveat needs to be discussed in a 'Limitations of the study' section at the end of the Discussion. This section also needs to discuss the fact that this study only uses one cell line for its experiments.

Response & Change:

We thank the reviewer for highlighting this important aspect. Indeed, we observed X chromosome bridge formation in the first generation of FuVis system (Fig. 1E in doi: 10.26508/lsa.202000911). Following reviewer 1's suggestion, we expanded our qPCR analysis using mCitrine-positive cells and confirmed that a panel of ISGs is not upregulated (Fig. 4M, S5A), and no signs of senescence were detected in these cells (Fig. S5B, C). Because mCitrine-positive cells contain not only MN, but also chromosome bridges and acentric chromosome fragments, these results imply these types of chromosome abnormalities do not robustly induce the cGAS-STING pathway in the context of our study. This interpretation has been incorporated into the manuscript (lines 369-372).

Nonetheless, we acknowledge the possibility that such chromosome abnormalities might activate cGAS-STING pathway under different cellular conditions. This consideration has been added to a newly created "limitations of the study" section (lines 418-424).

• There are some discrepancies in data presented that should be clarified or fixed. The experiments in Fig. 1B provide different results than those in Fig. S1H, where the same cell line and methods were used to generate the data. For example, in Fig. 1B, a maximum of 6% mCerulean3-positive cells were observed 9 days post-transduction of Cas9-sgF21 whereas ~20% were observed at day 9 in Fig. S1H. Is there a reason for the large variation in fluorescent cells?

Response & Change:

We thank the reviewer's comment. We noted that HEK293T cells were used for lentivirus production in the Fig. 1B experiment, whereas for the Fig. S1H experiment, we utilized HEK293T-derived Lenti-X cells (Takara Bio), which are known to yield higher viral titers. We have consistently observed more robust mCitrine and mCerulean3

expression with viruses generated from Lenti-X cells. We have clarified this methodological detail in the method section (lines 481, 490-491).

Similarly, the percentage of abnormal X chromosomes in the FuVis2.0 SC29 clone is listed as 6.7% in Fig. S1B but is shown as an average of >10% in Fig. S1C. Are the values listed in Fig. S1B representative of a single replicate? If so, it should be clarified.

Response & Change:

This is because result in Fig. S1B was an independent single experiment for the screening purpose. We have clarified this in the legend.

The authors also mention in lines 145-146 that neither mCitrine- or mCerulean3-positive cells had micronuclei in the first interphase but then indicate in lines 163-164 that both mCitrine- and mCerulean3-positive populations had micronuclei in the first interphase. The plot in Fig. 2C demonstrates that there are micronuclei present during the first interphase in both cell types, agreeing with the statement in lines 163-164.

Response & Change:

We agree that there are background levels of MN-positive cells and clarified this point in lines 147-149.

Finally, in line 207, it is stated that cGAS accumulates in ~50% of MN in the subsequent interphase but the bar plot in Fig. S3D indicates that cGAS accumulates in ~30%. These issues should be clarified or adjusted accordingly in the text.

Response & Change:

We meant that 50% of MN positive cells (cells with mCitrine and/or mScarlet-cGAS positive cytosolic foci). We have clarified this point (line 209).

• **Although the effect of the nucleosome binding mutant of cGAS on micronuclear DNA localization is clear in Fig. 3C, it is hard to interpret the negative results from the cGAS phospho-mutants since these experiments were performed in cells expressing endogenous cGAS. The experiments should be repeated following cGAS knockdown or knockout to assess the impact of the mutants on cGAS localization during mitosis. The expression of the cGAS mutants should also be evaluated by immunoblot analysis to compare to endogenous cGAS protein levels.**

Response & Change:

We have conducted KD experiments on cGAS in cells expressing shRNA-resistant cGAS-20A and cGAS-20DE. The results suggest that neither 20A nor 20DE impact the occurrence of the m-CAM event within the shcGAS background. Additionally, we have performed immunoblotting to assess the expression level of mScarlet-cGAS and the endogenous cGAS (Fig. S3A, D, E). These new results are described in lines 221-223.

• **Given the interesting observation that cGAS primarily binds to micronuclear chromosomes in mitosis, it would be a nice way to visualize the findings by extending the bars in Fig. 3A to include cGAS localization upon nuclear envelope breakdown (i.e., extending past timepoint '0'). It is also not too clear why the 'end of the imaging' category is included in the analyses in Fig. 3A. It would help to clarify the rationale for it.**

Response & Change:

We updated the Fig. 3A in line with the reviewer's suggestion. Our initial aim was to ascertain the frequency of the i-CAM (interphase cGAS accumulation) event. For this, we analyzed all MN-positive cells during the interphase, encompassing both those that transitioned into mitosis and those that did not within our observation period. Notably, three MN-positive cells exhibiting i-CAM did not proceed to mitosis during the limited duration of our imaging. To ensure a comprehensive representation of the i-CAM event and mitigate underestimation, we have included the "end of the imaging" category in this analysis. To clarify our focus on the interphase in this experiment, we have labeled "MN-positive interphase" above the bars in Fig. 3A.

- The authors should provide confirmation of the degree of H3K79me2 loss following seven days of DOT1L inhibitor treatment for the experiments performed in Fig. 3F. Although H3K79me2 reduction by DOT1L inhibitor has been confirmed in PMID: 36732527, those experiments were not performed in HCT116 cells and a direct comparison to interpret the experimental results is needed.

Response & Change:

We performed the IF analysis on H3K79me2, as well as H3K27me3 as a control. We confirmed reduction of H3K79me2 after 1-week DOT1L inhibitor treatment as reported (Fig. 3G and S3F).

Additional points

- A parental HCT116 control sample is missing from Fig. S1D.

Response & Change:

We have added a parental HCT116 control to Fig. S1D.

- Line 222: H3K79me2 is not a MN-specific histone modification. It is also found in the primary nucleus at sites of active transcription. The text should be adjusted accordingly.

Response & Change:

We have rephrased the text (line 225).

- Line 358: The authors should also cite PMID: 37286600 for epigenetic abnormalities in micronuclei.

Response & Change:

We have added the citation.

- The order of the panels in some figures can be confusing since they aren't presented in alphabetical order. For example, Fig. S3 has panel D at the top right rather than after panel C. Similar is found in Fig. 3, Fig. S1, and Fig. S4. It would help to re-organize the figure panels so that they appear in the order they are

referred to in the text.

Response & Change:

We have re-organized the figures 3, S1, and S4

There are some grammatical errors/typos that should be addressed.

Response & Change:

We appreciate the correction of grammatical errors/typos and have addressed them accordingly.

- **'Micronucleus' and 'micronuclei' are used interchangeably throughout the manuscript as the abbreviation 'MN' that was initially defined as 'micronucleus'. 'MNs' is also used as the plural of MN, which should be micronuclei not micronucleus'. It might be worth changing the MN abbreviation to 'micronuclei' throughout the text.**

Response & Change:

We have changed the MN abbreviation to micronuclei.

- **The abbreviation 'PN' is introduced twice in the manuscript in lines 35 and 174. The second instance should be removed and can just be referred to as PN.**

Response & Change:

We have removed the second introduction.

- **Lines 50-51: 'which phosphorylates TBK1 itself' could be re-worded to 'which then phosphorylates itself'.**

Response & Change:

We have re-worded the text.

- **Line 51: 'the' should be added before 'interferon regulatory factor 3'.**

Response & Change:

We have modified the text.

- **Lines 65 and 103: 'N -terminus' should be 'N-terminal'.**

Response & Change:

We have modified the text.

- **Line 70: 'different' can be removed.**

Response & Change:

We have modified the text.

- **Lines 72-73: 'This raises questions about how cGAS can be efficiently activated by MN in the presence of suppressive chromatin-cGAS interaction, with some studies suggesting that....' should be rephrased to 'This raises the question of how cGAS can be efficiently activated by MN in the presence of suppressive chromatin-cGAS interaction, with a recent study suggesting that....' since a single question and reference were provided.**

Response & Change:

We have modified the text.

- **Line 82: 'have' should be removed.**

Response & Change:

We have modified the text.

- **Line 84: 'has' should be removed.**

Response & Change:

We have modified the text.

- **Line 87: 'opportunity' can be re-worded to 'system'.**

Response & Change:

We have re-worded the text.

- **Line 129: 'chromosome' should be removed.**

Response & Change:

We have modified the text.

- **Line 146: 'fluorescent-positive' can be re-worded to 'fluorescent'.**

Response & Change:

We have re-worded the text.

- **Lines 156-167: 'analyzed from day 2 to 6 using a flow cytometer confirming...' should be rephrased to 'analyzed from day 2 to 6 using flow cytometry confirming...'.**

Response & Change:

We have re-worded the text.

- **Line 165: 'did' should be re-worded to 'do'.**

Response & Change:

We have re-worded the text.

- **Line 170: 'for' should be added before 'cells'.**

Response & Change:

We have added the text.

- **Line 213: 'the' should be added before 'cGASR236A-R255E mutant'.**

Response & Change:

We have added the text.

- **Line 217: 'phosphor' should be 'phospho'.**

Response & Change:

We have re-worded the text.

- **Line 219: 'This result indicates that....' can be re-worded to 'These results indicate that...'**

Response & Change:

We have re-worded the text.

- **Line 226: nucleosome should be plural.**

Response & Change:

We have re-worded the text.

- **The header in line 228 should be 'm-CAM does not lead to STING activation'.**

Response & Change:

We have re-worded the text.

- **Lines 247, 273, 315, 326, and 329: 'cxcl10' should be capitalized and italicized.**

Response & Change:

We have modified the text.

January 19, 2024

RE: Life Science Alliance Manuscript #LSA-2023-02424R

Dr. Makoto T Hayashi
Kyoto University
The Graduate School of Medicine
Yoshida-Konoe-cho
Sakyo-ku
Kyoto, Kyoto 60608501
Japan

Dear Dr. Hayashi,

Thank you for submitting your revised manuscript entitled "Micronucleus Is Not a Potent Inducer of cGAS-STING Pathway". We would be happy to publish your paper in Life Science Alliance pending final revisions necessary to meet our formatting guidelines.

- please address Reviewer 2's remaining comments
- please be sure that the authorship listing and order is correct
- please complete the Data Availability statement at this time
- please incorporate the Supplementary information and methods sections into the main manuscript. We do not have a size limit for the Materials and Methods section.

A. FINAL FILES:

B. MANUSCRIPT ORGANIZATION AND FORMATTING:

Sincerely,

Reviewer #1 (Comments to the Authors (Required)):

Dear authors, dear editors,

The manuscript "Micronucleus Is Not a Potent Inducer of cGAS-STING Pathway" by Dr. Makoto T Hayashi [manuscript #LSA-2023-02424R] has been now revised by authors.

To my view they addressed well all issues and questions raised by the other reviewer and myself. Authors also performed experiments that go beyond the scope of the manuscript to satisfy my comments. It is a strong and interesting work for the community and I do recommend its publication.

Best regards

Reviewer #2 (Comments to the Authors (Required)):

The authors have sufficiently addressed our concerns with new data and by re-wording the manuscript for clarity. In particular, the Limitations section is an important addition that now discusses aspects of the current study that will be essential to explore in future investigations. We appreciate the authors' attempts to isolate mCitrine-positive cells in the first mitosis to assess X chromosome abnormalities and understand the obstacles associated with doing so. Although we agree that the nature of the reporter does not affect the paper's conclusion on cGAS activation, defining how the reporter system works will be important for those who are interested in using it as an experimental tool and should be prioritized for future studies. We acknowledge that the authors have now noted the possibility of non-SCF genomic aberrations leading to mCitrine expression in the Limitations section to address this. We now just have a couple of minor comments regarding the revisions that should be clarified or adjusted before the manuscript is ready for publication.

Expressing the cGAS phospho-mutants in cGAS-depleted cells has now strengthened the claim that cGAS phosphorylation is not required for micronuclear DNA localization. However, it is unclear why there are three bands for mScarlet-cGAS in the confirmatory immunoblots in Figures S3A and S3D. This should be clarified, especially since there is a single band for endogenous cGAS.

Line 225: There should be a 'the' before histone modification.

Line 1079: There should be an 'of' after 1-week, and a 'the' before indicated doses.

We would like to thank the editor and reviewers again for handling our manuscript. We have highlighted changes made in response to **Reviewer 2** and the **Editor**'s comments in distinct colors. As required by the editor, the Data Availability statement has been completed. Additionally, the Supplementary information and method sections have been incorporated into the Materials & Methods section of the main manuscript.

Reviewer #2 (Comments to the Authors (Required)):

The authors have sufficiently addressed our concerns with new data and by re-wording the manuscript for clarity. In particular, the Limitations section is an important addition that now discusses aspects of the current study that will be essential to explore in future investigations. We appreciate the authors' attempts to isolate mCitrine-positive cells in the first mitosis to assess X chromosome abnormalities and understand the obstacles associated with doing so. Although we agree that the nature of the reporter does not affect the paper's conclusion on cGAS activation, defining how the reporter system works will be important for those who are interested in using it as an experimental tool and should be prioritized for future studies. We acknowledge that the authors have now noted the possibility of non-SCF genomic aberrations leading to mCitrine expression in the Limitations section to address this. We now just have a couple of minor comments regarding the revisions that should be clarified or adjusted before the manuscript is ready for publication.

Expressing the cGAS phospho-mutants in cGAS-depleted cells has now strengthened the claim that cGAS phosphorylation is not required for micronuclear DNA localization. However, it is unclear why there are three bands for mScarlet-cGAS in the confirmatory immunoblots in Figures S3A and S3D. This should be clarified, especially since there is a single band for endogenous cGAS.

Response & Change:

We thank the reviewer for raising this point. Our interpretation is as follows:

1. **Top Band:** We utilized a blastR (blasticidin resistant gene)-p2a-mScarlet-cGAS construct, where p2a is a self-cleaving peptide. Given that the efficiency of self-cleaving is not 100%, the top weak band likely corresponds to the

blast-R-p2a-mScarlet-cGAS peptide, approximately 105 kDa in size.

- Middle Band:** This band corresponds to the p2a-self-cleaved mScarlet-cGAS peptide, approximately 91 kDa in size.
- Bottom Band:** It is reported that RFPs, including DsRed and its derivatives, undergo cleavage during maturation. For example, in DsRed, the cleavage site is between [Phe65] and [Gln66-Tyr67-Gly68] (refer to Fig. 1B in [https://doi.org/10.1016/S1097-2765\(03\)00393-9](https://doi.org/10.1016/S1097-2765(03)00393-9)). mScarlet, developed from an artificially synthesized mRed7 (based on mCherry and other natural RFPs, as detailed in <https://doi.org/10.1038/nmeth.4074>), is likely to undergo similar cleavage (10.5281/zenodo.6450250). In mScarlet, we identified Phe66-Met67-Tyr68-Gly69, analogous to DsRed's cleavage site. Thus, we suppose that a peptide bond between Phe66 and Met67 is cleaved, resulting in an approximately 82 kDa mScarlet(67-236)-3FL-cGAS peptide. Indeed, purified mScarlet has been reported to show two cleaved fragments [9 kDa and 19 kDa] ("Characterization of purity of 6xHis-mScarlet preparation" in https://andrewgyork.github.io/mScarlet_lifetime_reports_pH/appendix.html). We clarified these interpretation in the Materials & Methods section (lines 680-686).

The following figure represents our interpretation.

Line 225: There should be a 'the' before histone modification.

Line 1079: There should be an 'of' after 1-week, and a 'the' before indicated doses.

Response & Change:

We thank the reviewer and have amended the text.

January 22, 2024

RE: Life Science Alliance Manuscript #LSA-2023-02424RR

Dr. Makoto T Hayashi
Kyoto University
The Graduate School of Medicine
Yoshida-Konoe-cho
Sakyo-ku
Kyoto, Kyoto 60608501
Japan

Dear Dr. Hayashi,

Thank you for submitting your Research Article entitled "Micronucleus Is Not a Potent Inducer of cGAS-STING Pathway". It is a pleasure to let you know that your manuscript is now accepted for publication in Life Science Alliance. Congratulations on this interesting work.

DISTRIBUTION OF MATERIALS:

Again, congratulations on a very nice paper. I hope you found the review process to be constructive and are pleased with how the manuscript was handled editorially. We look forward to future exciting submissions from your lab.

Sincerely,
